# From Compression to Generalization: Language Model Distillation With Grafting

## Abstract

Knowledge distillation is a widely used technique in distilling large language models. It is applied both for strong-to-weak distillation, where large-scale flagship models serve as teachers to produce lightweight models suitable for deployment, and for weak-to-strong distillation, where previous-generation models contribute to the development of stronger next-generation models. From a model compression perspective of knowledge distillation, students may be encouraged to adopt mode-seeking behavior; however, for building generalizable generative language models, mode-covering behavior should also be considered. To address this, we conduct an experimental analysis and propose a simple yet effective *grafting* strategy, in which sequence trees generated at multiple temperatures for autoregressive modeling are combined into a single distillation target. Our extensive experiments demonstrate the effectiveness of the proposed grafting approach.

## 1 Introduction

The deep learning era, which commenced around 2010 (Sevilla et al., 2022), has been characterized by a significant expansion in the computational scale of neural network architectures. Concurrently, *knowledge distillation* (Hinton et al., 2015) has emerged as a prominent *model compression* technique, wherein a smaller student model is trained to emulate the behavior of a larger, more powerful teacher model. This strategy remains highly relevant and effective for contemporary large language models, exemplified by GPT-3 (Brown et al., 2020), which are typically composed of billions of parameters. In practice, public large language models are often deployed in distilled forms, consisting of a few to tens of billions of parameters and derived from main models composed of hundreds of billions of parameters (Yang et al., 2025; Guo et al., 2025). As such, knowledge distillation has emerged as a standard technique for machine learning, even in the era of large language models.

A key distinction of knowledge distillation in generative language models, compared to the discriminative setting originally introduced by Hinton et al. (2015), is that teacher-generated outputs can be directly used as training data. In fact, simply fine-tuning the student on these samples is effectively equivalent to minimizing the forward Kullback–Leibler (KL; Kullback & Leibler, 1951) divergence between the teacher and student distributions (Kim & Rush, 2016). From a model compression perspective, this process can be seen as a lower-capacity student imitating only a subset of the teacher's modes, producing generations that are fluent but less diverse (Cha & Cho, 2025). It suggests that, rather than utilizing the *mode-covering* forward KL, the *mode-seeking* reverse KL may be more appropriate. Recent approaches have adopted this perspective by incorporating reverse KL divergence terms into their loss formulations (Gu et al., 2024; Agarwal et al., 2024; Ko et al., 2025).

While knowledge distillation was originally proposed for model compression, continued research has revealed that its role extends beyond this simple function. Notably, Stanton et al. (2021) showed that a highly generalizable student does not necessarily possess high distillation fidelity, suggesting the student is not simply a compressed version of the teacher. It implies that the distillation itself contributes to the training of generalizable models, suggesting its function goes beyond simple compression (Yim et al., 2017; Furlanello et al., 2018; Xie et al., 2020). In this context, *weak-to-strong distillation* has been considered for language models, exploring how an existing, weaker teacher model can help develop the next generation of stronger models (Qin et al., 2022; Lee et al., 2023). Indeed, the DeepSeek series utilizes knowledge distillation to train a new generation of models by leveraging the knowledge of its predecessors (Shirgaonkar et al., 2024; Guo et al., 2025).

The main contributions of our work can be summarized as follows:

- Adopting a more up-to-date perspective that views knowledge distillation as more than a tool for model compression, we question the appropriateness of guiding generative language model distillation purely toward mode-seeking behavior (Section 3). Although such an approach may be suitable for compression, it can limit the student's ability to capture the full diversity and complexity of language, thereby constraining generalization.

- Even state-of-the-art algorithms that employ the mode-seeking reverse KL often mix it with the mode-covering forward KL, suggesting that blindly enforcing mode-seeking should be avoided and that balancing mode-covering and mode-seeking behaviors is crucial for successful distillation. Building on this insight, we propose a simple yet effective strategy, *grafting*, which enables the student to better incorporate mode-covering behavior.

- The proposed grafting approach is a simple yet effective method that combines sequence trees generated at multiple temperatures to create the distillation target (Section 4). It is clearly distinct from merely aggregating multi-temperature samples and is broadly applicable across existing algorithms, as shown by extensive experiments in both strong-to-weak and weak-to-strong distillation scenarios for large language models (Section 5).

## 2 RELATED WORK

**Knowledge distillation.** The idea of training a smaller model to imitate the behavior of a larger one has been explored under the name *model compression* (Buciluă et al., 2006). The seminal work of Hinton et al. (2015) popularized this approach for neural networks, introducing the term *knowledge distillation*. Since then, it has proven effective across diverse subfields of machine learning, including computer vision (Chen et al., 2017), speech recognition (Chebotar & Waters, 2016), natural language processing (Sanh et al., 2019), and reinforcement learning (Rusu et al., 2016), and has become a widely adopted technique. Alongside these developments, analyses have highlighted that the benefits of knowledge distillation extend far beyond simple model compression (Yim et al., 2017; Cho & Hariharan, 2019; Yuan et al., 2020; Stanton et al., 2021; Allen-Zhu & Li, 2023).

**Distillation in modern large language models.** In recent advancements in large language model development, knowledge distillation has become a de facto standard technique, not only for training smaller models but also as part of training large models themselves (Grattafiori et al., 2024; Abdin et al., 2024; Team et al., 2025; Yang et al., 2025). Unlike distillation in discriminative models (Hinton et al., 2015), where the student learns to mimic the teacher's outputs on a given dataset, distillation in autoregressive language models can be performed by training the student on sequences generated by the teacher (Kim & Rush, 2016). Consequently, it does not necessarily require an explicit dataset; rather, the use of synthetic data or annotations produced by existing models effectively serves as a practical form of distillation (Shirgaonkar et al., 2024; Guo et al., 2025).

**Distillation from weak teacher.** Although distillation was originally introduced in the context of model compression (Hinton et al., 2015), its benefits are not limited to cases where the teacher model is larger. Early works in computer vision have shown that even students of equal or greater size can benefit from distillation (Furlanello et al., 2018; Yuan et al., 2020; Xie et al., 2020; Stanton et al., 2021). In the context of language models, it raises the question of *weak-to-strong distillation*: how an existing model, acting as a *weak* teacher, can help develop the next generation of *stronger* models (Qin et al., 2022; Lee et al., 2023). Notably, the training of modern large language models already incorporates synthetic data generated by previous generations of models (Liu et al., 2024; Guo et al., 2025), actually serving as a form of distillation from a weak teacher.

**Temperature in knowledge distillation.** Since the seminal work of Hinton et al. (2015), applying temperature scaling to categorical targets has become standard practice in knowledge distillation. While it can be interpreted as interpolating between logit and label matching (Kim et al., 2021), its effectiveness is highly sensitive to careful tuning, with no universal guideline. To address this, several studies in discriminative modeling setups have explored adaptively adjusting the temperature (Liu et al., 2022; Li et al., 2023b; Sun et al., 2024) or employing multiple temperatures simultaneously (Jin et al., 2023; Chi et al., 2023). However, to the best of our knowledge, its role in generative language model distillation remains largely unexplored, with only a few studies addressing it, e.g., experimentally in Ouyang et al. (2024) and theoretically in Cha & Cho (2025).

# 3 PATHOLOGIES OF REDUCED DIVERSITY IN DISTILLATION

## 3.1 PRELIMINARIES

*Autoregressive language modeling* assumes that the true distribution $p_*$ over a sequence $s_{1:T}$ conditioned on an initial token $s_0$, along with corresponding vocabulary $\mathcal{V}_*$, can be factorized as

$$p_*(s_{1:T}|s_0) = \prod_{t=1}^{T} p_*(s_t|s_{0:t-1}), \text{ where } \forall t : s_t \in \mathcal{V}_*. \tag{1}$$

Given observations assumed to be drawn from $p_*$, and a finite vocabulary $\mathcal{V} \subset \mathcal{V}_*$ constructed accordingly, we define a statistical model $p_\theta$, parameterized by $\theta$. The model is trained to empirically minimize $D_{\mathrm{KL}}(p_*\|p_\theta)$ estimated from data, and the resulting $p_\theta$ is thereby expected to represent Equation 1. In practice, it is often modified to yield sharper outputs for more fluent generation; for instance, a common approach is *temperature scaling* (Ackley et al., 1985):

$$p_{\theta,\tau}(s_t = v|s_{0:t-1}) = \frac{\exp\left(\log p_\theta(s_t = v|s_{0:t-1})/\tau\right)}{\sum_{w \in \mathcal{V}} \exp\left(\log p_\theta(s_t = w|s_{0:t-1})/\tau\right)}, \text{ with } \tau \in (0, \infty). \tag{2}$$

Modern autoregressive language modeling has successfully leveraged large-scale data and neural network to effectively model $p_\theta$ (Brown et al., 2020). Once such a model $p_\theta$ is obtained, it can serve as a teacher for *knowledge distillation* to efficiently train a new model $p_\phi$. Since sampling from the teacher $p_\theta$ is fully tractable, distillation can be readily carried out. For instance, sequence-level knowledge distillation (SeqKD; Kim & Rush, 2016) minimizes the expected negative log-likelihood $\mathbb{E}_{s_{1:T} \sim p_\theta(\cdot|s_0)}[-\log p_\phi(s_{1:T}|s_0)] = \mathbb{E}_{s_{1:T} \sim p_\theta(\cdot|s_0)}[-\sum_{t=1}^{T} \log p_\phi(s_t|s_{0:t-1})]$ as an empirical approximation of the KL divergence $D_{\mathrm{KL}}(p_\theta\|p_\phi)$.

## 3.2 MODE-COVERING VERSUS MODE-SEEKING

Discussions about the *mode-covering* nature of the forward KL divergence persist in the distillation of autoregressive language models. Specifically, minimizing $D_{\mathrm{KL}}(p_\theta\|p_\phi)$ encourages the student model's categorical distribution $p_\phi(s_t|s_{0:t-1})$ to spread its probability mass widely over the vocabulary $\mathcal{V}$. When the student model's capacity is limited, this can lead to the *mode-averaging* problem, where the learned distribution fails to capture any distinct mode of the teacher distribution. Given that an autoregressive language model should assign high probability to at least one plausible next token at each step to generate fluent output, an overly smooth $p_\phi(s_t|s_{0:t-1})$ would be undesirable.

Several recent studies have addressed this issue by replacing the forward KL divergence with alternatives, such as: *1) the reverse KL divergence* (Gu et al., 2024) $D_{\mathrm{RKL}}(p_\theta, p_\phi) = D_{\mathrm{KL}}(p_\phi\|p_\theta)$, *2) the generalized Jensen-Shannon divergence* (Agarwal et al., 2024) $D_{\mathrm{JSD},\beta}(p_\theta, p_\phi) = \beta D_{\mathrm{KL}}(p_\theta\|\beta p_\theta + (1-\beta)p_\phi) + (1-\beta)D_{\mathrm{KL}}(p_\phi\|\beta p_\theta + (1-\beta)p_\phi)$ with $\beta \in [0,1]$, and *3) the skew forward and reverse KL divergences* (Ko et al., 2024) $D_{\mathrm{SKL},\alpha}(p_\theta, p_\phi) = D_{\mathrm{KL}}(p_\theta\|\alpha p_\theta + (1-\alpha)p_\phi)$ and $D_{\mathrm{SRKL},\alpha}(p_\theta, p_\phi) = D_{\mathrm{KL}}(p_\phi\|\alpha p_\phi + (1-\alpha)p_\theta)$ with $\alpha \in [0,1]$. The shared philosophy behind these approaches is to leverage the *mode-seeking* nature of the reverse KL divergence, encouraging the student to better capture individual modes rather than covering or averaging multiple modes.

More recently, Cha & Cho (2025) provide a minimal working explanation of knowledge distillation in generative models by analyzing Gaussian mixture models: $p_*(x) = \sum_{k=1}^{K_*} \alpha_{*,k} \mathcal{N}(x; \mu_{*,k}, \Sigma_{*,k})$, $p_\theta(x) = \sum_{k=1}^{K_\theta} \alpha_{\theta,k} \mathcal{N}(x; \mu_{\theta,k}, \Sigma_{\theta,k})$, and $p_\phi(x) = \sum_{k=1}^{K_\phi} \alpha_{\phi,k} \mathcal{N}(x; \mu_{\phi,k}, \Sigma_{\phi,k})$. The $\tau$-modulated teacher, $p_{\theta,\tau}(x) = \sum_{k=1}^{K_\theta} \alpha_{\theta,k,\tau} \mathcal{N}(x; \mu_{\theta,k}, \Sigma_{\theta,k})$, with temperature-scaled mixture coefficients $\alpha_{\theta,k,\tau} = \alpha_{\theta,k}^{1/\tau} / \sum_{j=1}^{K_\theta} \alpha_{\theta,j}^{1/\tau}$, yields a distillation objective bounded by Jensen's inequality:

$$\int p_{\theta,\tau}(x) \log p_\phi(x)\mathrm{d}x \geq \sum_{k=1}^{K_\theta} \sum_{k'=1}^{K_\phi} \alpha_{\theta,k,\tau} \alpha_{\phi,k'} \int \mathcal{N}(x; \mu_{\theta,k}, \Sigma_{\theta,k}) \log \mathcal{N}(x; \mu_{\phi,k'}, \Sigma_{\phi,k'})\mathrm{d}x \tag{3}$$

Intuitively, lowering $\tau$ makes $\alpha_{\theta,k,\tau}$ sparse, i.e., most of the coefficients become near zero, so the student selectively captures only the high-weight modes rather than covering the full distribution. This naturally encourages mode-seeking behavior, which is particularly pronounced for a limited-capacity student ($K_\phi \ll K_\theta$). Their strong-to-weak distillation simulation using Gaussian mixtures shows that training a student to generate high-quality samples requires lowering $\tau$ to encourage mode-seeking during the distillation procedure, a principle they extend to language models where quality is prioritized over diversity, such as in instruction tuning or downstream generation tasks.

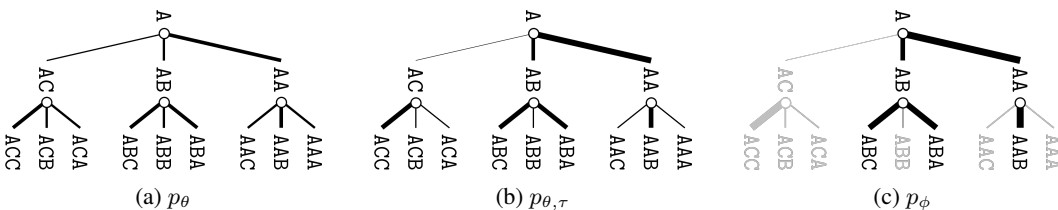

$$(a)\ p_\theta \qquad\qquad (b)\ p_{\theta,\tau} \qquad\qquad (c)\ p_\phi$$

Figure 1: Conceptual illustration of sequence modeling for $\mathcal{V} = \{\mathtt{A}, \mathtt{B}, \mathtt{C}\}$. Line thickness represents the transition probability over possible next tokens, i.e., $p(s_t = v | s_{0:t-1})$ for $v \in \mathcal{V}$.

### 3.3 DIVERSITY VERSUS SIMPLICITY

At each time step in left-to-right generation in Equation 1, $p_*(s_t | s_{0:t-1})$ can be viewed as a mixture coefficient over the possible next tokens $v \in \mathcal{V}_*$, thereby inducing a mixture distribution over the possible continuations $s_{t+1:T}$. Specifically, the distribution over $s_{t+1:T}$ given $s_{0:t-1}$ factorizes as

$$p_*(s_{t+1:T} | s_{0:t-1}) = \sum_{v \in \mathcal{V}_*} p_*(s_t = v | s_{0:t-1}) \cdot p_*(s_{t+1:T} | s_t = v, s_{0:t-1}). \tag{4}$$

In other words, at each time step $t$, the generation process resembles a branching point with $|\mathcal{V}_*|$ possible limbs extending forward. In the context of distillation, encouraging the student model to adopt mode-seeking behavior corresponds to pruning weaker branches while allowing stronger ones to grow. For instance, making the teacher model $p_\theta$ more selective by applying temperature scaling with $\tau < 1$ effectively retains only the high-probability branches while pruning the others:

$$\sum_{v \in \mathcal{V}} p_{\theta,\tau}(s_t = v | s_{0:t-1}) \cdot p_{\theta,\tau}(s_{t+1:T} | s_t = v, s_{0:t-1}). \tag{5}$$

As $\tau \to 0$, an increasing number of components are pruned, i.e., $p_{\theta,\tau}(s_t = v | s_{0:t-1}) \approx 0$, thereby simplifying the student model $p_\phi$'s task to modeling only $K \ll |\mathcal{V}|$ active branches:

$$\sum_{k=1}^{K} \alpha_{t,k} \cdot p_\phi(s_{t+1:T} | s_t = v_k, s_{0:t-1}), \quad \text{where } \alpha_{t,k} = \frac{p_\phi(s_t = v_k | s_{0:t-1})}{\sum_{j=1}^{K} p_\phi(s_t = v_j | s_{0:t-1})}, \tag{6}$$

and $\{v_k\}_{k=1}^{K}$ denotes the set of surviving branches after pruning. Reducing $\tau$ thus narrows the range of continuations the student model considers, *simplifying* likelihood modeling and making learning easier for the student. Figure 1 depicts a conceptual illustration of sequence modeling at each stage of distillation: (a) the original teacher model $p_\theta$, (b) the selective teacher model $p_{\theta,\tau}$ with $\tau < 1$, and (c) the student model $p_\phi$ distilled from $p_{\theta,\tau}$. Here, the sequence tree that the student is required to model becomes simplified, which effectively reduces the complexity of the learning task.

However, it is worth questioning whether this simplification is truly desirable. The success of modern large language models, extending even to complex reasoning capabilities, stems from their capacity to predict the next token across an enormous and diverse set of possible continuations (Radford et al., 2018; 2019; Brown et al., 2020). Namely, their strength lies in modeling the likelihood over a highly complex and richly branching sequence tree. While simplifying this tree by focusing only on a reduced set of high-probability branches during the distillation procedure may be appropriate from the perspective of *model compression*, it fundamentally risks limiting the model's ability to capture the full diversity and complexity of language, potentially restricting its performance and generalization capacity. From a more up-to-date perspective that treats distillation as more than mere model compression (Stanton et al., 2021), such simplification is unlikely to be optimal. Indeed, state-of-the-art algorithms employ the reverse KL divergence, but rather than fully replacing forward KL with reverse KL, they use a mixture of the two (Agarwal et al., 2024; Ko et al., 2025), highlighting that blindly encouraging the student model to be mode-seeking is not the optimal strategy.

### 3.4 EMPIRICAL ANALYSIS

In this section, we build on the experimental setup for autoregressive language models of Cha & Cho (2025) to examine precision and recall as the student model capacity varies; precision measures the extent to which samples drawn from the model $p_\phi$ are likely under $p_*$, i.e., mode-seeking, whereas recall measures how comprehensively the model $p_\phi$ covers the modes of $p_*$, i.e., mode-covering:

$$\begin{aligned} \text{Precision}(p_\phi) &= \mathbb{E}_{s_{1:T} \sim p_\phi(\cdot | s_0)} \left[ \log p_*(s_{1:T} | s_0) \right], \\ \text{Recall}(p_\phi) &= \mathbb{E}_{s_{1:T} \sim p_*(\cdot | s_0)} \left[ \log p_\phi(s_{1:T} | s_0) \right]. \end{aligned} \tag{7}$$

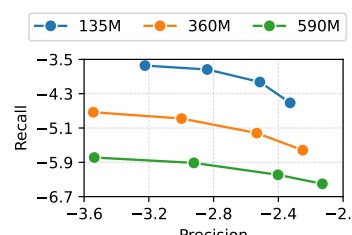

Figure 3: Conceptual illustration of the *grafted* sequence tree provided by a teacher model $p_{\theta,\tau}$ for $\mathcal{V} = \{\texttt{A}, \texttt{B}, \texttt{C}\}$ and $\mathcal{T} = \{1.0, 2.0, 3.0\}$. Line thickness represents the transition probability over possible next tokens, i.e., $p_{\theta,\tau}(s_t = v \mid s_{0:t-1})$ for $v \in \mathcal{V}$ and $\tau \in \mathcal{T}$.

**Setup.** For analysis, we treat the pre-trained SmolLM2-1.7B model (Allal et al., 2025) as the true distribution $p_*$. We first train a 360M teacher model $p_\theta$ from scratch by minimizing $D_{\mathrm{KL}}(p_* \| p_\theta)$ on one million sequences sampled from $p_*$, each with a maximum length of $T = 100$ and starting token $s_0 = \texttt{The}$. The 135M (smaller), 360M (equal), and 590M (larger) student models $p_\phi$ are then trained from scratch by minimizing $D_{\mathrm{KL}}(p_\theta \| p_\phi)$ on one million sequences generated by the teacher model $p_\theta$, using the same sequence length and starting token $p_\theta$. To vary the teacher's selectivity during distillation, we apply temperature scaling with $\tau \in \{0.7, 1.0, 1.5, 2.0\}$ and use min-$p = 0.1$ sampling (Minh et al., 2025) to avoid degenerate outputs at higher temperatures. Further experimental details are provided in Appendix B.1.

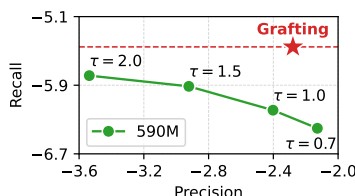

Figure 2: Precision-recall curve.

**Results.** Figure 2 shows that decreasing $\tau$, thereby making the teacher more selective, induces a consistent precision–recall trade-off (Cha & Cho, 2025) across all student model scales: precision increases while recall decreases (cf. —●—; where the rightmost points correspond to $\tau = 0.7$). Notably, as the student size increases (cf. 135M → 360M → 590M), recall decreases consistently across all $\tau$ values. This observation indicates that larger students, once capable of faithfully imitating the teacher's simplified sequence tree, tend to fail in adequately covering the true data distribution.

## 4 DIVERSIFYING SEQUENCE TREE VIA GRAFTING

### 4.1 MOTIVATION

In Section 3.4, we observed that simply increasing $\tau$ to make the teacher's sequence tree less selective is insufficient to improve the recall of the larger student while preserving its precision, as the student model is still capable to overfit the simplified sequence tree provided by teacher. While learning the teacher-induced simplified language modeling may ease training and align with a model compression perspective, it does not achieve the more fundamental goal of developing a truly generalizable language model. This limitation is particularly problematic in weak-to-strong distillation scenarios, where the objective is to build a stronger next-generation model from a weaker predecessor. Even in strong-to-weak distillation settings, the challenge would remain pronounced, since modern large language model pipelines often equip the *weak* student with *billions of parameters*, making it questionable to describe such a student as a truly *limited-capacity* model.

Figure 4: Grafting results.

### 4.2 APPROACH

To address this, we propose a simple yet effective technique, *grafting*, which combines sequence trees generated at multiple temperatures in the set $\mathcal{T}$:

$$q_\theta(s_{1:T}|s_0) = \sum_{\tau \in \mathcal{T}} \pi_0(\tau) p_{\theta,\tau}(s_{1:T}|s_0), \text{ where } \forall \tau \in \mathcal{T} : \pi_0(\tau) = \frac{1}{|\mathcal{T}|}. \tag{8}$$

Figure 3 illustrates the diversified sequence tree produced by grafting, and Figure 4 presents proof-of-concept results extending the empirical analysis in Section 3.4, showing that grafting effectively improves recall (cf. ★). It is worth noting that this improvement is not simply due to aggregating sequences generated at multiple temperatures into a larger dataset; the *same* number of one million sequences drawn from the grafted sequence tree with $\mathcal{T} = \{0.7, 1.0, 1.5, 2.0\}$ to train the student. Compared with distillation from a non-grafted sequence tree (cf. —•—), grafting yields a substantial recall increase while preserving high precision. The result showcases that simply grafting sequence trees across multiple temperatures is an effective way to ensure diversity of the distillation target, thereby facilitating the training of a more generalizable language model.

*Remark.* It should be noted that these results are obtained at a relatively small scale using a synthetic true data distribution. The recall improvement from grafting for the 560M student remains modest compared with that of the 135M student (cf. Figures 2 and 4), which we attribute to the simplicity of the task and the limited capacity of the models. Accordingly, these experiments should be regarded primarily as motivating evidence of grafting's potential for distilling generative language models. In the following sections, we therefore assess its effectiveness in both weak-to-strong and strong-to-weak distillation settings under more realistic scenarios involving billion-parameter models.

## 4.3 IMPLEMENTATION

As demonstrated in our proof-of-concept experiments, implementing the grafting approach for the SeqKD (Kim & Rush, 2016) algorithm is straightforward: sample $\tau \sim \pi_0$, roll out $s_{1:T} \sim p_{\theta,\tau}$, and train the student by minimizing negative log-likelihood on those sequences:

$$\mathbb{E}_{\tau \sim \pi_0} \mathbb{E}_{s_{1:T} \sim p_{\theta,\tau}(\cdot|s_0)} \left[ -\sum_{t=1}^{T} \log p_\phi(s_t|s_{0:t-1}) \right]. \tag{9}$$

However, recent state-of-the-art algorithms, such as generalized knowledge distillation (GKD; Agarwal et al., 2024) and DistiLLM (Ko et al., 2024; 2025), explicitly compute divergence terms for given sequences as in vanilla knowledge distillation (KD; Hinton et al., 2015). To faithfully target the grafted mixture in such cases, we should consider the forward and reverse KL terms

$$\mathcal{L}_{\mathrm{fKL}}(\phi) = \mathbb{E}_{s_{1:T} \sim q_\theta(\cdot|s_0)} \left[ \sum_{t=1}^{T} D_{\mathrm{KL}}(q_\theta(\cdot|s_{0:t-1})\|p_\phi(\cdot|s_{0:t-1})) \right],$$

$$\mathcal{L}_{\mathrm{rKL}}(\phi) = \mathbb{E}_{s_{1:T} \sim q_\theta(\cdot|s_0)} \left[ \sum_{t=1}^{T} D_{\mathrm{KL}}(p_\phi(\cdot|s_{0:t-1})\|q_\theta(\cdot|s_{0:t-1})) \right], \tag{10}$$

with the token-level marginal and posterior:

$$q_\theta(s_t|s_{0:t-1}) = \sum_{\tau \in \mathcal{T}} \pi_t(\tau|s_{0:t-1}) p_{\theta,\tau}(s_t|s_{0:t-1}),$$

$$\pi_t(\tau|s_{0:t-1}) = \frac{\pi_0(\tau)p_{\theta,\tau}(s_{1:t-1}|s_0)}{\sum_{\tau'} \pi_0(\tau')p_{\theta,\tau'}(s_{1:t-1}|s_0)}, \quad \pi_0(\tau) = \frac{1}{|\mathcal{T}|}. \tag{11}$$

Since computing equation 10 is not tractable, we optimize tractable upper-bounds computed from sequences obtained by first sampling $\tau \sim \pi_0$ and then rolling out $s_{1:T} \sim p_{\theta,\tau}(\cdot \mid s_0)$:

$$\hat{\mathcal{L}}_{\mathrm{fKL}}(\phi) = \mathbb{E}_{\tau \sim \pi_0} \mathbb{E}_{s_{1:T} \sim p_{\theta,\tau}(\cdot|s_0)} \left[ \sum_{t=1}^{T} D_{\mathrm{KL}}(p_{\theta,\tau}(\cdot|s_{0:t-1})\|p_\phi(\cdot|s_{0:t-1})) \right],$$

$$\hat{\mathcal{L}}_{\mathrm{rKL}}(\phi) = \mathbb{E}_{\tau \sim \pi_0} \mathbb{E}_{s_{1:T} \sim p_{\theta,\tau}(\cdot|s_0)} \left[ \sum_{t=1}^{T} D_{\mathrm{KL}}(p_\phi(\cdot|s_{0:t-1})\|p_{\theta,\tau}(\cdot|s_{0:t-1})) \right]. \tag{12}$$

These upper bounds share the same gradient as the original losses with respect to $\phi$.

*Proof.* Appendix A provides detailed derivations and further extends the results to mixed sources, for example $\alpha q_\theta + (1 - \alpha)p_\phi$, which we consider since both GKD and DistiLLM are implemented in this way. Shortly, for the forward KL case, we show that $\nabla_\phi \mathcal{L}_{\mathrm{fKL}}(\phi) = \nabla_\phi \hat{\mathcal{L}}_{\mathrm{fKL}}(\phi)$ by matching the gradient contributions at each time step $t$. For the reverse KL case, we start from

$$\hat{\mathcal{L}}_{\mathrm{rKL}}(\phi) = \mathbb{E}_{s_{1:T} \sim q_\theta(\cdot|s_0)} \sum_{t=1}^{T} \mathbb{E}_{\tau \sim \pi_t(\cdot|s_{0:t-1})} \left[ D_{\mathrm{KL}}(p_\phi(\cdot|s_{0:t-1})\|p_{\theta,\tau}(\cdot|s_{0:t-1})) \right], \tag{13}$$

which follows from the identity $q_\theta(s_{1:t-1}|s_0)\pi_t(\tau|s_{0:t-1}) = \pi_0(\tau)p_{\theta,\tau}(s_{1:t-1}|s_0)$. Comparing the time-$t$ contributions of $\mathcal{L}_{\mathrm{rKL}}$ and $\hat{\mathcal{L}}_{\mathrm{rKL}}$, we show that $\mathcal{L}_{\mathrm{rKL}}(\phi) \leq \hat{\mathcal{L}}_{\mathrm{rKL}}(\phi)$.

*Remark on computational costs.* Training sequences can be generated with this implementation in mind without any conflict with recent high-throughput inference stacks such as vLLM (Kwon et al.,

Table 1: Main results for general-purpose instruction-following benchmarks using Qwen2.5 models: winning rates in percentile on AlpacaEval (AE), Evol-Inst (EI), and UltraFeed (UF).

| | 3B-Instruct ($p_\theta$) → 1.5B ($p_\phi$) | | | | 3B-Instruct ($p_\theta$) → 7B ($p_\phi$) | | | |
|---|---|---|---|---|---|---|---|---|
| Method | AE | EI | UF | Avg. | AE | EI | UF | Avg. |
| Teacher ($p_\theta$) | 65.4 | 60.9 | 58.0 | 61.4 | 65.4 | 60.9 | 58.0 | 61.4 |
| Student ($p_\phi$) | 38.3 | 17.2 | 24.6 | 26.7 | 56.1 | 29.0 | 36.3 | 40.5 |
| SeqKD (Kim & Rush, 2016) | 42.8 | 43.0 | 43.1 | 43.0 | 64.2 | 62.6 | 54.9 | 60.6 |
| *w/ grafting (ours)* | 46.2 | 43.4 | 42.5 | **44.0** | 65.2 | 64.5 | 55.4 | **61.7** |
| KD (Hinton et al., 2015) | 43.5 | 41.9 | 42.2 | 42.5 | 62.4 | 63.7 | 54.0 | 60.0 |
| *w/ grafting (ours)* | 48.1 | 45.1 | 42.6 | **45.3** | 64.0 | 64.7 | 55.1 | **61.3** |
| GKD (Agarwal et al., 2024) | 48.8 | 47.3 | 46.5 | 47.5 | 66.4 | 64.2 | 55.4 | 62.0 |
| *w/ grafting (ours)* | 48.3 | 48.2 | 48.0 | **48.2** | 70.0 | 68.0 | 56.8 | **64.9** |
| DistiLLM-2 (Ko et al., 2025) | 47.1 | 46.5 | 44.6 | 46.1 | 65.8 | 64.9 | 55.4 | 62.0 |
| *w/ grafting (ours)* | 49.6 | 47.1 | 45.5 | **47.4** | 68.5 | 67.5 | 59.0 | **65.0** |

2023), since it only requires adding an extra temperature column. At each training iteration, only $p_{\theta,\tau}$ for a single $\tau$ is needed, which is directly derived from $p_\theta$ and $\tau$. Thus, pre-generating $s_{1:T}$ and storing $\tau$ as part of the dataset, like as Ko et al. (2025), remains a valid and effective strategy.

*Remark on mixture weights.* One can use a weighted mixture instead of the uniform mixture we currently employ for $\pi_0$ in Equation 8. However, we defaulted to the uniform prior for both theoretical and practical reasons. In Equation 11, $\pi_0$ can be viewed as the prior and $\pi_t$ as the posterior. Since $\pi_0$ reflects the state at time $t = 0$ where we have no information regarding how good each temperature's subtree is, it is more rational to employ a noninformative uniform prior, even though an engineered weighted mixture might potentially yield superior results. Such an engineered, non-uniform prior might exist, this is not a principled prior and adds engineering complexity to the algorithm. Moreover, we observe that $\pi_t$ is conditioned on the path traversed on the grafted sequence tree up to that point in Equation 11. This structure effectively constitutes a posterior update based on preceding observations from the initial uniform prior at $t = 0$, rendering the non-uniform mixture at $t > 0$.

## 5 EXPERIMENTS

We empirically validate the effectiveness of our proposed grafting approach on general-purpose instruction-following tasks and further conduct analyses including ablation studies, scalability evaluation, and architecture generalization (Section 5.1). We then extend our evaluation to other tasks, covering mathematical reasoning and code generation (Section 5.1). Appendix B provides detailed experimental settings as well as supplementary materials. It is worth noting that the total number of training sequences was maintained consistently across both the baseline (without grafting) and the grafted results to ensure a fair comparison throughout our experiments.

### 5.1 GENERAL-PURPOSE INSTRUCTION-FOLLOWING

**Setup.** We evaluate our approach on general-purpose instruction-following tasks using the Qwen2.5 family (Qwen et al., 2024). The pre-trained 3B-Instruct model is used as the teacher, and the 1.5B and 7B base models are fine-tuned via knowledge distillation. Training is performed on 50k prompts sampled from UltraChat200k (Ding et al., 2023), and evaluation is conducted on AlpacaEval (Li et al., 2023a), Evol-Instruct (Xu et al., 2024), and UltraFeedback (Cui et al., 2024). For LLM-as-a-Judge evaluation (Zheng et al., 2023), we use gpt-5-nano-2025-08-07 as the judge, with text-davinci-003 serving as the rival for AlpacaEval and gpt-3.5-turbo for Evol-Instruct and UltraFeedback. Our setup largely follows Ko et al. (2025), with further details provided in Appendix B.2.

**Main results.** Table 1 presents results for both strong-to-weak (3B-Instruct → 1.5B) and weak-to-strong (3B-Instruct → 7B) distillation setups. Our proposed grafting approach consistently improves upon standard baselines such as SeqKD (Kim & Rush, 2016) and KD (Hinton et al., 2015), as well as state-of-the-art methods including GKD (Agarwal et al., 2024) and DistiLLM-2 (Ko et al., 2025).

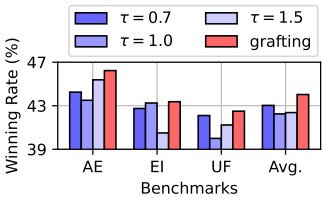 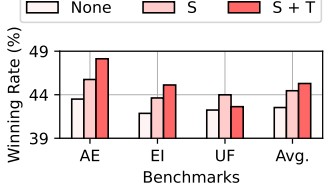 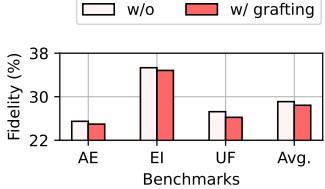

| (a) Efficacy of grafting. | (b) Ablation on implementation. | (c) Fidelity analysis. |

Figure 5: Ablation results for (a) comparing grafted and single trees, (b) distilling with a grafted tree target, and (c) assessing distillation fidelity.

These results highlight the broad applicability of grafting and reinforce our earlier argument that, in realistic large language model distillation scenarios involving students with billions of parameters, it is important to account for both mode-seeking and mode-covering behaviors rather than focusing solely on the former, in both strong-to-weak and weak-to-strong setups.

Notably, the average improvement in the strong-to-weak setup is 1.45%p (2.63% of the possible maximum gain), while the weak-to-strong setup achieves a larger improvement of 2.08%p (5.34% of the possible maximum gain). In the weak-to-strong setup, grafting not only allows the student to perform on par with the weak teacher using only SeqKD ($60.6 \rightarrow 61.7 \approx 61.4$) or KD ($60.0 \rightarrow 61.3 \approx 61.4$), but also complements GKD ($62.0 \rightarrow 64.9 \gg 61.4$) and DistiLLM-2 ($62.0 \rightarrow 65.0 \gg 61.4$), enabling the student to substantially outperform the teacher. This indicates that grafting's benefits in both mode-covering and mode-seeking become more pronounced as the student's capacity increases, while also complementing the mode-seeking tendencies of state-of-the-art algorithms.

**Scaling to larger student.** Table 2 further evaluates the scalability of our approach by distilling to the 32B base model (SeqKD; 3B-Instruct $\rightarrow$ 32B). This weak-to-strong distillation setup, in which the student's parameter count exceeds that of the teacher by over an order of magnitude, not only demonstrates the scalability of our method but also reinforces our central claim that mode-covering becomes increasingly important as the student's capacity grows. The results show that our proposed grafting ap-

Table 2: Qwen2.5 3B-Instruct $\rightarrow$ 32B.

| Method | AE | EI | UF | Avg. |
|---|---|---|---|---|
| Teacher | 65.4 | 60.9 | 58.0 | 61.4 |
| Student | 69.1 | 40.5 | 40.4 | 50.0 |
| SeqKD | 70.3 | 64.9 | 58.2 | 64.4 |
| *w/ grafting* | 71.4 | 65.8 | 60.5 | **65.9** |

proach (i) consistently outperforms baselines, underscoring its strong potential to scale effectively to larger models, and (ii) achieves an improvement of 1.48%p (4.16% of the possible maximum gain), further confirming the increasing benefits of mode-covering as the student's capacity grows.

**Extending to another model family.** To further demonstrate that the proposed grafting approach is not tied to a specific model family, we extend our evaluation to the Llama3 models (Grattafiori et al., 2024), using Llama3.1 8B as the student and Llama3.2 3B-Instruct as the teacher. As shown in Table 3, grafting consistently improves upon the baseline in this weak-to-strong distillation setup, confirming that its benefits generalize beyond the Qwen2.5 models examined in the main results. Notably, while the

Table 3: Llama3 3B-Instruct $\rightarrow$ 8B.

| Method | AE | EI | UF | Avg. |
|---|---|---|---|---|
| Teacher | 71.4 | 60.4 | 56.0 | 62.6 |
| Student | 69.8 | 39.9 | 49.5 | 53.1 |
| SeqKD | 68.9 | 41.1 | 51.7 | 53.9 |
| *w/ grafting* | 70.5 | 41.4 | 53.3 | **55.1** |

student already attains competitive performance on AlpacaEval and vanilla SeqKD fails to deliver further gains, grafting still improves upon the baseline, demonstrating its robustness and effectiveness even in challenging scenarios where conventional distillation schemes fall short.

**Comparing grafted and single trees.** Although Table 1 already reports baseline results under a single tree setting, namely the hyperparameters from Ko et al. (2025): top-$p = 0.95$ sampling (Holtzman et al., 2020) with $\tau = 0.8$, here we further compare gainst the individual sub-trees that constitute our grafted tree. Figure 5a shows the distillation results (SeqKD; 3B-Instruct $\rightarrow$ 1.5B) when using a single sub-tree at each temperature value (0.7, 1.0, and 1.5) with min-$p = 0.1$ sampling (Minh et al., 2025), alongside the grafted tree. The results make it clear that combining them into a grafted tree yields better performance and thus validate the efficacy of our proposed grafting approach.

Table 4: Results for mathematical reasoning benchmarks using Qwen2.5 and Qwen2.5-Math models: Pass@1 on MathQA (MQA), GSM8K (GSM), and MinervaMath (MNV).

| | 1.5B ($p_\phi$) | | | | 7B ($p_\phi$) | | | | 7B-Math ($p_\phi$) | | | |
|---|---|---|---|---|---|---|---|---|---|---|---|---|
| Method | MQA | GSM | MNV | Avg. | MQA | GSM | MNV | Avg. | MQA | GSM | MNV | Avg. |
| Teacher ($p_\theta$) | 45.0 | 27.3 | 42.0 | 38.1 | 45.0 | 27.3 | 42.0 | 38.1 | 45.0 | 27.3 | 42.0 | 38.1 |
| Student ($p_\phi$) | 35.0 | 8.8 | 27.1 | 23.6 | 43.4 | 11.3 | 39.1 | 31.3 | 55.4 | 37.7 | 45.8 | 46.3 |
| DistiLLM-2 | 37.4 | 40.9 | 30.1 | 36.1 | 47.9 | 46.7 | 40.1 | 44.9 | 56.1 | 39.0 | 45.6 | 46.9 |
| *w/ grafting (ours)* | 37.7 | 42.6 | 31.2 | **37.2** | 48.8 | 46.9 | 40.2 | **45.3** | 56.4 | 40.3 | 46.3 | **47.7** |

Table 5: Results for code generation benchmarks using DeepSeek-Coder models: Pass@1 on HumanEval (HE), HumanEval+ (HE+), MBPP (MB), and MBPP+ (MB+).

| | 6.7B-Instruct ($p_\theta$) → 1.3B ($p_\phi$) | | | | | 1.3B-Instruct ($p_\theta$) → 6.7B ($p_\phi$) | | | | |
|---|---|---|---|---|---|---|---|---|---|---|
| Method | HE | HE+ | MB | MB+ | Avg. | HE | HE+ | MB | MB+ | Avg. |
| Teacher ($p_\theta$) | 76.8 | 72.6 | 75.1 | 65.6 | 72.5 | 64.6 | 61.6 | 62.4 | 52.1 | 60.2 |
| Student ($p_\phi$) | 33.5 | 28.1 | 57.4 | 49.2 | 42.1 | 48.8 | 41.5 | 73.0 | 57.4 | 55.2 |
| DistiLLM-2 (Ko et al., 2025) | 39.0 | 32.9 | 61.6 | 51.1 | 46.2 | 50.0 | 45.7 | 68.8 | 58.2 | 55.7 |
| *w/ grafting (ours)* | 41.5 | 37.2 | 61.6 | 50.8 | **47.8** | 51.2 | 45.7 | 71.2 | 59.0 | **56.8** |

**Distilling with a grafted tree target.** As discussed in Section 4.3, methods that explicitly compute token-level KL terms should properly treat the grafted tree as the distillation target. We conduct ablation on this point in Figure 5b: 'None' denotes the baseline without grafting (KD; 3B-Instruct → 1.5B). 'S' uses the grafted sequences only as training data, while 'S + T' both uses the grafted sequences as training data and follows the implementation described in Section 4.3. The results show that using the grafted tree as the target consistently improves performance, while also demonstrating that grafting is clearly distinct from simply sampling training sequences at multiple temperatures.

**Assessing distillation fidelity.** Our approach views distillation not merely as model compression, that is, producing a high-fidelity student, but as a process for obtaining a more generalizable model. Following Stanton et al. (2021), we assess distillation fidelity to verify that the performance gains from grafting are not simply due to closer alignment with the teacher; in fact, fidelity can be slightly lower, since the target has been replaced by the grafted tree. To this end, we introduce a new fidelity metric that normalizes the average predictive KL against that of a non-distilled student: a score of 100% indicates predictions completely same as the teacher, while 0% corresponds to the same level of fidelity as the non-distilled student (see Appendix B.2 for the precise definition). Figure 5c shows that the distillation results (SeqKD; 3B-Instruct → 1.5B) with grafting do not yield higher fidelity than those without grafting, as expected.

## 5.2 MATHEMATICAL REASONING AND CODE GENERATION

**Setup.** We further evaluate our approach on mathematical reasoning and code generation tasks. For the mathematical reasoning experiments, we utilize the Qwen2.5 and Qwen2.5-Math families (Qwen et al., 2024; Yang et al., 2024). For the code generation experiments, we use the DeepSeek-Coder family (Guo et al., 2024). Appendices B.3 and B.4 provide further experimental details.

**Results for mathematical reasoning.** Table 4 presents DistiLLM-2 (Ko et al., 2025) results, which show consistent gains achieved through grafting. Here, the teacher is the 1.5B-Instruct model, while the students are the 1.5B and 7B models. Training utilizes 50k samples from MetaMathQA (Yu et al., 2024), and evaluation is performed on MathQA (Amini et al., 2019), GSM8K (Cobbe et al., 2021), and MinervaMath (Lewkowycz et al., 2022) via the Language Model Evaluation Harness (Gao et al., 2024). Moreover, Table 6 confirms this consistent improvement on the more challenging AMC23. Appendix B.3 extends the 1.5B results from Table 4 to further reasoning-specialized model (Guo

Table 6: Results for AMC23.

| Method | 1.5B | 7B | 7B-Math |
|---|---|---|---|
| Teacher | 12/40 | 12/40 | 12/40 |
| Student | 1/40 | 6/40 | 12/40 |
| DistiLLM-2 | 10/40 | 18/40 | 12/40 |
| *w/ grafting* | **11/40** | **21/40** | **14/40** |

et al., 2025), showcasing the potential scope of grafting on more optimized, state-of-the-art reasoning models. Furthermore, Appendix B.3 presents a failure case for the 0.5B student, supporting our observation that grafting is more effective when the student possesses sufficient capacity.

**Results for code generation.** Table 5 presents DistiLLM-2 (Ko et al., 2025) results, where grafting yields performance improvements across both strong-to-weak and weak-to-strong distillation setups. All models are sourced from the DeepSeek-Coder family, with the strong-to-weak setup using the 6.7B-Instruct model to teach the 1.3B base model, and the weak-to-strong setup using the 1.3B-Instruct model to teach the 6.7B base model. Training uses Evol-Instruct-Code-80k-v1 (Roshdieh, 2023), an open-source implementation of Code Evol-Instruct (Luo et al., 2023), and evaluation is conducted on HumanEval (Chen et al., 2021) and MBPP (Austin et al., 2021) using EvalPlus (Liu et al., 2023), which additionally includes the extended benchmarks HumanEval+ and MBPP+.

**Temperature range and diminishing returns.** Unless otherwise specified, our grafting results utilize a grafted target constructed with temperatures $\tau \in \{0.7, 1.0, 1.5\}$, a range selected to maintain reasonable generation quality under min-$p$ sampling (Minh et al., 2025). While higher temperatures could enhance mode-covering behavior, we hypothesized that excessively high temperatures would degrade the quality of base generations, thereby diminishing the benefits of grafting. To validate this, we evaluated performance by extending the temperature candidates to include values exceeding typical ranges. Specifically, we compared our standard set against expanded sets including $\tau = 2.0$ and $\tau = 3.0$, using the mathematical reasoning setup (DistiLLM-2; 1.5B-Math-Instruct $\rightarrow$ 1.5B). As anticipated, Table 7 indicates a clear performance drop at these higher values, identifying a point of diminishing returns. This suggests that our grafting approach relies on plausible generation candidates produced at standard temperature levels. Consequently, the adopted set of $\{0.7, 1.0, 1.5\}$ represents a robust choice grounded in common practice, rather than the result of extensive hyperparameter tuning.

Table 7: Ablation on temperature range.

| Method | MQA | GSM | MNV | Avg. |
|---|---|---|---|---|
| Teacher | 45.0 | 27.3 | 42.0 | 38.1 |
| Student | 35.0 | 8.8 | 27.1 | 23.6 |
| DistiLLM-2 | 37.4 | 40.9 | 30.1 | 36.1 |
| *w/ grafting (1.5)* | 37.7 | 42.6 | 31.2 | **37.2** |
| *w/ grafting (2.0)* | 37.1 | 42.7 | 29.8 | 36.5 |
| *w/ grafting (3.0)* | 36.9 | 42.5 | 29.3 | 36.2 |

## 6 CONCLUSION

We revisited knowledge distillation for autoregressive generative language models, emphasizing that its role extends beyond mere model compression. While purely mode-seeking approaches may be suitable from the model compression perspective, they can limit the student's ability to capture linguistic diversity and hinder generalization in language modeling. To address this, we introduce *grafting*, a simple yet effective strategy that merges sequence trees generated at multiple temperatures, balancing mode-seeking and mode-covering behaviors. Extensive experiments demonstrate its broad applicability across both classical and recent algorithms, as well as in strong-to-weak and weak-to-strong distillation scenarios. Our findings suggest that successful distillation requires careful consideration of both tendencies, and that grafting provides a practical and generalizable tool for improving language model distillation as models scale in size and complexity.

## ETHICS STATEMENT

This study makes use of publicly available large language models and datasets, which may inherently reflect and potentially amplify societal biases present in their training corpora. To mitigate associated ethical concerns, we provide a transparent account of the libraries, models, and datasets employed, the details of which are provided in the Appendix, explicitly state their original licenses, and comply with the corresponding codes of conduct.

## REPRODUCIBILITY STATEMENT

Appendix A provides the complete proofs for the discussions in Section 4.3. Appendix B details the experimental setup, including the datasets, model architectures, pretrained checkpoints and their licenses, specific hyperparameters used for training, the libraries and codebases employed, as well as the hardware configurations.

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

## A PROOFS AND DERIVATIONS

### A.1 FORWARD AND REVERSE KL CASES

We show $\nabla_\phi \mathcal{L}_{\text{fKL}}(\phi) = \nabla_\phi \hat{\mathcal{L}}_{\text{fKL}}(\phi)$. For time $t$, the gradient contribution of $\mathcal{L}_{\text{fKL}}$ is

$$
\begin{aligned}
&\mathbb{E}_{s_{1:t-1}\sim q_\theta(\cdot|s_0)}\mathbb{E}_{s_t\sim q_\theta(\cdot|s_{0:t-1})}\left[-\nabla_\phi\log p_\phi(s_t|s_{0:t-1})\right]\\
&= \mathbb{E}_{s_{1:t-1}\sim q_\theta(\cdot|s_0)}\sum_{\tau\in\mathcal{T}}\pi_t(\tau|s_{0:t-1})\mathbb{E}_{s_t\sim p_{\theta,\tau}(\cdot|s_{0:t-1})}\left[-\nabla_\phi\log p_\phi(s_t|s_{0:t-1})\right]\\
&= \int q_\theta(s_{1:t-1}|s_0)\sum_{\tau\in\mathcal{T}}\pi_t(\tau|s_{0:t-1})\mathbb{E}_{s_t\sim p_{\theta,\tau}(\cdot|s_{0:t-1})}\left[-\nabla_\phi\log p_\phi(s_t|s_{0:t-1})\right]\mathrm{d}s_{1:t-1}\\
&= \int q_\theta(s_{1:t-1}|s_0)\sum_{\tau\in\mathcal{T}}\frac{\pi_0(\tau)p_{\theta,\tau}(s_{1:t-1}|s_0)}{q_\theta(s_{1:t-1}|s_0)}\mathbb{E}_{s_t\sim p_{\theta,\tau}(\cdot|s_{0:t-1})}\left[-\nabla_\phi\log p_\phi(s_t|s_{0:t-1})\right]\mathrm{d}s_{1:t-1}\\
&= \int\sum_{\tau\in\mathcal{T}}\pi_0(\tau)p_{\theta,\tau}(s_{1:t-1}|s_0)\mathbb{E}_{s_t\sim p_{\theta,\tau}(\cdot|s_{0:t-1})}\left[-\nabla_\phi\log p_\phi(s_t|s_{0:t-1})\right]\mathrm{d}s_{1:t-1},
\end{aligned}
\tag{14}
$$

which matches the time-$t$ gradient contribution of $\hat{\mathcal{L}}_{\text{fKL}}$:

$$
\begin{aligned}
&\mathbb{E}_{\tau\sim\pi_0}\mathbb{E}_{s_{1:t-1}\sim p_{\theta,\tau}(\cdot|s_0)}\mathbb{E}_{s_t\sim p_{\theta,\tau}(\cdot|s_{0:t-1})}\left[-\nabla_\phi\log p_\phi(s_t|s_{0:t-1})\right]\\
&= \sum_{\tau\in\mathcal{T}}\pi_0(\tau)\int p_{\theta,\tau}(s_{1:t-1}|s_0)\mathbb{E}_{s_t\sim p_{\theta,\tau}(\cdot|s_{0:t-1})}\left[-\nabla_\phi\log p_\phi(s_t|s_{0:t-1})\right]\mathrm{d}s_{1:t-1}.
\end{aligned}
\tag{15}
$$

Next, we show $\mathcal{L}_{\text{rKL}} \leq \hat{\mathcal{L}}_{\text{rKL}}$. Using the identity $q_\theta(s_{1:t-1}|s_0)\pi_t(\tau|s_{0:t-1}) = \pi_0(\tau)p_{\theta,\tau}(s_{1:t-1}|s_0)$,

$$
\hat{\mathcal{L}}_{\text{rKL}}(\phi) = \mathbb{E}_{s_{1:T}\sim q_\theta(\cdot|s_0)}\sum_{t=1}^{T}\mathbb{E}_{\tau\sim\pi_t(\cdot|s_{0:t-1})}\left[D_{\text{KL}}(p_\phi(\cdot|s_{0:t-1})||p_{\theta,\tau}(\cdot|s_{0:t-1}))\right].
\tag{16}
$$

For time $t$, the loss contribution of $\hat{\mathcal{L}}_{\text{rKL}}$ is

$$
\begin{aligned}
\hat{\mathcal{L}}_{\text{rKL}}^{(t)}(\phi) &= \mathbb{E}_{\tau\sim\pi_t(\cdot|s_{0:t-1})}\left[D_{\text{KL}}(p_\phi(\cdot|s_{0:t-1})||p_{\theta,\tau}(\cdot|s_{0:t-1}))\right]\\
&= \mathbb{E}_{\tau\sim\pi_t(\cdot|s_{0:t-1})}\left[\mathbb{E}_{s_t\sim p_\phi(\cdot|s_{0:t-1})}\left[\log p_\phi(s_t|s_{0:t-1}) - \log p_{\theta,\tau}(s_t|s_{0:t-1})\right]\right]\\
&= \mathbb{E}_{s_t\sim p_\phi(\cdot|s_{0:t-1})}\left[\mathbb{E}_{\tau\sim\pi_t(\cdot|s_{0:t-1})}\left[\log p_\phi(s_t|s_{0:t-1}) - \log p_{\theta,\tau}(s_t|s_{0:t-1})\right]\right]\\
&= \mathbb{E}_{s_t\sim p_\phi(\cdot|s_{0:t-1})}\left[\log p_\phi(s_t|s_{0:t-1})\right] - \mathbb{E}_{s_t\sim p_\phi(\cdot|s_{0:t-1})}\left[\mathbb{E}_{\tau\sim\pi_t(\cdot|s_{0:t-1})}\left[\log p_{\theta,\tau}(s_t|s_{0:t-1})\right]\right].
\end{aligned}
\tag{17}
$$

Meanwhile, the loss contribution of $\mathcal{L}_{\text{rKL}}$ is

$$
\begin{aligned}
\mathcal{L}_{\text{rKL}}^{(t)}(\phi) &= D_{\text{KL}}(p_\phi(\cdot|s_{0:t-1})||q_\theta(\cdot|s_{0:t-1}))\\
&= \mathbb{E}_{s_t\sim p_\phi(\cdot|s_{0:t-1})}\left[\log p_\phi(s_t|s_{0:t-1}) - \log q_\theta(s_t|s_{0:t-1})\right]\\
&= \mathbb{E}_{s_t\sim p_\phi(\cdot|s_{0:t-1})}\left[\log p_\phi(s_t|s_{0:t-1})\right] - \mathbb{E}_{s_t\sim p_\phi(\cdot|s_{0:t-1})}\left[\log q_\theta(s_t|s_{0:t-1})\right].
\end{aligned}
\tag{18}
$$

By Jensen's inequality,

$$
\begin{aligned}
\mathbb{E}_{s_t\sim p_\phi(\cdot|s_{0:t-1})}\left[\log q_\theta(s_t|s_{0:t-1})\right] &= \mathbb{E}_{s_t\sim p_\phi(\cdot|s_{0:t-1})}\left[\log \mathbb{E}_{\tau\sim\pi_t(\cdot|s_{0:t-1})}\left[p_{\theta,\tau}(s_t|s_{0:t-1})\right]\right]\\
&\geq \mathbb{E}_{s_t\sim p_\phi(\cdot|s_{0:t-1})}\left[\mathbb{E}_{\tau\sim\pi_t(\cdot|s_{0:t-1})}\left[\log p_{\theta,\tau}(s_t|s_{0:t-1})\right]\right],
\end{aligned}
\tag{19}
$$

hence $\mathcal{L}_{\text{rKL}}^{(t)}(\phi) \leq \hat{\mathcal{L}}_{\text{rKL}}^{(t)}(\phi)$ for all $t$, and summing over $t$ gives $\mathcal{L}_{\text{rKL}}(\phi) \leq \hat{\mathcal{L}}_{\text{rKL}}(\phi)$. Since both $p_\theta$ and $p_\phi$ define token distributions via softmax over finite logits in practice, (a) exchanging $\nabla_\phi$ with $\mathbb{E}$ in the forward-KL case is justified, and (b) the Jensen step above is valid with finite expectations.

### A.2 SKEWED FORWARD AND REVERSE KL CASES

We show $\mathcal{L}_{\text{SKL},\alpha} \leq \hat{\mathcal{L}}_{\text{SKL},\alpha}$ for the forward case, where

$$
\begin{aligned}
\mathcal{L}_{\text{SKL},\alpha}(\phi) &= \mathbb{E}_{s_{1:T}\sim q_\theta(\cdot|s_0)}\left[\sum_{t=1}^{T}D_{\text{SKL},\alpha}(q_\theta(\cdot|s_{0:t-1}), p_\phi(\cdot|s_{0:t-1}))\right],\\
\hat{\mathcal{L}}_{\text{SKL},\alpha}(\phi) &= \mathbb{E}_{\tau\sim\pi_0}\mathbb{E}_{s_{1:T}\sim p_{\theta,\tau}(\cdot|s_0)}\left[\sum_{t=1}^{T}D_{\text{SKL},\alpha}(p_{\theta,\tau}(\cdot|s_{0:t-1}), p_\phi(\cdot|s_{0:t-1}))\right].
\end{aligned}
\tag{20}
$$

Using the identity $q_\theta(s_{1:t-1}|s_0)\pi_t(\tau|s_{0:t-1}) = \pi_0(\tau)p_{\theta,\tau}(s_{1:t-1}|s_0)$ and the joint convexity of the KL divergence, we derive the following for time $t$, yielding $\mathcal{L}_{\text{SKL},\alpha} \leq \hat{\mathcal{L}}_{\text{SKL},\alpha}$ by summing over $t$:

$$
\begin{aligned}
&D_{\text{SKL},\alpha}(q_\theta(\cdot|s_{0:t-1}), p_\phi(\cdot|s_{0:t-1}))\\
&= D_{\text{KL}}\left(\sum_\tau \pi_t(\tau|s_{0:t-1})p_{\theta,\tau}(\cdot|s_{0:t-1})||\sum_\tau \pi_t(\tau|s_{0:t-1})(\alpha p_{\theta,\tau}(\cdot|s_{0:t-1}) + (1-\alpha)p_\phi(\cdot|s_{0:t-1}))\right)\\
&\leq \sum_\tau \pi_t(\tau|s_{0:t-1})D_{\text{KL}}\left(p_{\theta,\tau}(\cdot|s_{0:t-1})||(\alpha p_{\theta,\tau}(\cdot|s_{0:t-1}) + (1-\alpha)p_\phi(\cdot|s_{0:t-1}))\right)\\
&= \sum_\tau \pi_t(\tau|s_{0:t-1})D_{\text{SKL},\alpha}(p_{\theta,\tau}(\cdot|s_{0:t-1}), p_\phi(\cdot|s_{0:t-1})).
\end{aligned}
\tag{21}
$$

Next, we show $\mathcal{L}_{\text{SRKL},\alpha} \leq \hat{\mathcal{L}}_{\text{SRKL},\alpha}$ for the reverse case, where

$$
\begin{aligned}
\mathcal{L}_{\text{SRKL},\alpha}(\phi) &= \mathbb{E}_{s_{1:T} \sim q_\theta(\cdot|s_0)} \left[ \textstyle\sum_{t=1}^T D_{\text{SKL},\alpha}(p_\phi(\cdot|s_{0:t-1}), q_\theta(\cdot|s_{0:t-1})) \right], \\
\hat{\mathcal{L}}_{\text{SRKL},\alpha}(\phi) &= \mathbb{E}_{\tau \sim \pi_0} \mathbb{E}_{s_{1:T} \sim p_{\theta,\tau}(\cdot|s_0)} \left[ \textstyle\sum_{t=1}^T D_{\text{SKL},\alpha}(p_\phi(\cdot|s_{0:t-1}), p_{\theta,\tau}(\cdot|s_{0:t-1})) \right].
\end{aligned}
\tag{22}
$$

Using the identity $q_\theta(s_{1:t-1}|s_0)\pi_t(\tau|s_{0:t-1}) = \pi_0(\tau)p_{\theta,\tau}(s_{1:t-1}|s_0)$ and the convexity of the KL divergence, we derive the following for time $t$, yielding $\mathcal{L}_{\text{SRKL},\alpha} \leq \hat{\mathcal{L}}_{\text{SRKL},\alpha}$ by summing over $t$:

$$
\begin{aligned}
&D_{\text{SKL},\alpha}\left(p_\phi(\cdot|s_{0:t-1}), q_\theta(\cdot|s_{0:t-1})\right) \\
&= D_{\text{KL}}\left(p_\phi(\cdot|s_{0:t-1}) \| \textstyle\sum_\tau \pi_t(\tau|s_{0:t-1})(\alpha p_\phi(\cdot|s_{0:t-1}) + (1-\alpha)p_{\theta,\tau}(\cdot|s_{0:t-1}))\right) \\
&\leq \textstyle\sum_\tau \pi_t(\tau|s_{0:t-1})D_{\text{KL}}\left(p_\phi(\cdot|s_{0:t-1}) \| \alpha p_\phi(\cdot|s_{0:t-1}) + (1-\alpha)p_{\theta,\tau}(\cdot|s_{0:t-1})\right) \\
&= \textstyle\sum_\tau \pi_t(\tau|s_{0:t-1})D_{\text{SRKL},\alpha}\left(p_\phi(\cdot|s_{0:t-1}), p_{\theta,\tau}(\cdot|s_{0:t-1})\right).
\end{aligned}
\tag{23}
$$

# B    EXPERIMENTAL DETAILS

The main deep learning libraries employed in our experiments are publicly available on GitHub, together with their respective licenses:

- DeepSpeed (deepspeedai/DeepSpeed; Apache-2.0 license)
- FlashAttention (Dao-AILab/flash-attention; BSD-3-Clause license)
- Hugging Face Accelerate (huggingface/accelerate; Apache-2.0 license)
- Hugging Face Datasets (huggingface/datasets; Apache-2.0 license)
- Hugging Face PEFT (huggingface/peft; Apache-2.0 license)
- Hugging Face Transformers (huggingface/transformers; Apache-2.0 license)
- Hugging Face TRL (huggingface/trl; Apache-2.0 license)
- PyTorch (pytorch/pytorch; BSD-3-Clause license)
- vLLM (vllm-project/vllm; Apache-2.0 license)

## B.1    PROOF-OF-CONCEPT EXPERIMENTS

**Model.** We provide the architectural details of the 135M, 360M, 590M, and 1.7B models used in the experiments of Sections 3.4 and 4. All models are based on the SmolLM2 configuration (Allal et al., 2025), with the 1.7B model utilizing a pre-trained checkpoint publicly available on the Hugging Face Hub (HuggingFaceTB/SmolLM2-1.7B), which is licensed under Apache 2.0.

| # Params | # Layers | Hidden size | Intermediate size | # Attention heads | # Key-value heads |
|---|---|---|---|---|---|
| 135M | 30 | 576 | 1535 | 9 | 3 |
| 360M | 32 | 960 | 2560 | 15 | 5 |
| 590M | 32 | 1024 | 4096 | 16 | 16 |
| 1.7B | 24 | 2048 | 8192 | 32 | 32 |

**Optimization.** We trained the models for 10 epochs using the Adam optimizer (Kingma & Ba, 2015) with hyperparameters $\beta_1 = 0.9$, $\beta_2 = 0.95$, $\epsilon = 10^{-8}$, and zero weight decay. The learning rate was set to $5 \times 10^{-4}$ with a linear scheduler and a warmup ratio of 0.1. Training was performed in mixed precision (FP16) using DeepSpeed with ZeRO Stage 2 optimization (Rajbhandari et al., 2020). We used a total batch size of 256 with gradient accumulation steps of 1 and gradient clipping set to 1.0. Depending on resource availability, training was conducted on either four NVIDIA RTX 3090 GPUs or four NVIDIA A6000 GPUs.

**Evaluation.** Precision and recall were computed empirically with 10,000 samples. Precision was measured by generating 10,000 sequences from the trained student model ($p_\phi$) and averaging their log-likelihoods under the 1.7B model ($p_*$). Recall was measured by generating 10,000 sequences from the 1.7B model ($p_*$) and averaging their log-likelihoods under the trained student model ($p_\phi$).

## B.2 GENERAL-PURPOSE INSTRUCTION-FOLLOWING EXPERIMENTS

**Model.** We employed the pre-trained Qwen2.5 checkpoints from Hugging Face Hub for our experiments, all of which are released under the Apache 2.0 license: base models (Qwen/Qwen2.5-1.5B; Qwen/Qwen2.5-3B; Qwen/Qwen2.5-7B; Qwen/Qwen2.5-32B) and instruction-tuned models (Qwen/Qwen2.5-0.5B-Instruct; Qwen/Qwen2.5-1.5B-Instruct; Qwen/Qwen2.5-3B-Instruct). In our additional experiments, we also used the pre-trained Llama3.1 and Llama3.2 checkpoints from Hugging Face Hub, which are released under the Llama3.1[1] and 3.2[2] Community License: base model (meta-llama/Llama-3.1-8B) and instruction-tuned model (meta-llama/Llama-3.2-3B-Instruct).

**Dataset.** We used UltraChat200k (Ding et al., 2023, MIT license) for training, and AlpacaEval (Li et al., 2023a, Apache 2.0 license), Evol-Instruct (Xu et al., 2024, CC BY-NC 4.0 license), and UltraFeedback (Cui et al., 2024) for evaluation, all in their preprocessed form from the official DistiLLM-2 (Ko et al., 2025) codebase, publicly available on GitHub under an unspecified license (jongwooko/distillm-2).

**Optimization.** Our experimental setup largely follows the configurations provided in the official DistiLLM-2 codebase (Ko et al., 2025), which is publicly available on GitHub under an unspecified license (jongwooko/distillm-2). As recommended in the codebase when the base model serves as the student, we first performed supervised fine-tuning before applying any distillation algorithms. Fine-tuning was carried out in bfloat16 precision using DeepSpeed with ZeRO Stage 3 optimization (Rajbhandari et al., 2020), with a total batch size of 512, for one epoch over the UltraChat200k dataset. We used the Adam optimizer (Kingma & Ba, 2015) with hyperparameters $\beta_1 = 0.9$, $\beta_2 = 0.999$, $\epsilon = 10^{-8}$, and zero weight decay. The learning rate was set to $5.0 \times 10^{-5}$ with cosine scheduling and a 10% warmup, and gradients were clipped at 1.0. Fine-tuning for Qwen2.5 1.5B and 7B models was performed on four NVIDIA A6000 GPUs, while Qwen2.5 32B and Llama3.1 8B models were trained on eight NVIDIA A6000 GPUs. For distillation, we applied LoRA (Hu et al., 2022) fine-tuning with rank 16, $\alpha = 128$, and dropout 0.05 across all attention and feed-forward projection matrices. Distillation was performed for one epoch on the UltraChat200k dataset, using the same optimization hyperparameters as above, except batch size of 32.

**Evaluation.** Following Ko et al. (2025), we adopted the same pairwise comparison prompt for LLM-as-a-Judge (Zheng et al., 2023) evaluation:

---

[System]
Please act as an impartial judge and evaluate the quality of the responses provided by two AI assistants to the user question displayed below. You should choose the assistant that follows the user's instructions and answers the user's question better. Your evaluation should consider factors such as the helpfulness, relevance, accuracy, depth, creativity, and level of detail of their responses. Begin your evaluation by comparing the two responses and provide a short explanation. Avoid any position biases and ensure that the order in which the responses were presented does not influence your decision. Do not allow the length of the responses to influence your evaluation. Do not favor certain names of the assistants. Be as objective as possible. After providing your explanation, output your final verdict by strictly following this format: "[[A]]" if assistant A is better, "[[B]]" if assistant B is better, and "[[C]]" for a tie.

[User Question]
{question}

[The Start of Assistant A's Answer]
{answer_a}
[The End of Assistant A's Answer]

[The Start of Assistant B's Answer]
{answer_b}
[The End of Assistant B's Answer]

---

[1]https://www.llama.com/llama3_1/license/
[2]https://www.llama.com/llama3_2/license/

Model outputs were generated using top-$p = 0.95$ sampling (Holtzman et al., 2020) with $\tau = 0.8$ and a maximum length of 512. The judge model, gpt-5-nano-2025-08-07, evaluated each model by comparing its outputs against the corresponding baseline (text-davinci-003 for AlpacaEval, and gpt-3.5-turbo for both Evol-Instruct and UltraFeedback) to calculate the winning rate. To mitigate position bias, results were averaged after swapping the order of the responses being compared.

**Fidelity metric.** Let $p_\theta$ be a teacher and $p_\phi$ be a student model. The average predictive KL divergence for quantifying the distillation fidelity is given by (Stanton et al., 2021)

$$\text{AvgPredKL}(\phi) = \tfrac{1}{T} \sum_{t=1}^{T} D_{\text{KL}}(p_\theta(\cdot|s_{0:t-1})||p_\phi(\cdot|s_{0:t-1})), \tag{24}$$

where lower values indicate higher fidelity. Based on this quantity, we define a normalized fidelity score between 0 and 1 by using the non-distilled student $p_{\phi_{\text{non-distilled}}}$ as a reference:

$$\text{Fidelity}(\phi_{\text{distilled}}) = 1 - \frac{\text{AvgPredKL}(\phi_{\text{distilled}})}{\text{AvgPredKL}(\phi_{\text{non-distilled}})} \tag{25}$$

Here, a value of 0 corresponds to the average predictive KL of an undistilled student at the same level, whereas a value of 1 indicates perfect alignment with the teacher.

### B.3 MATHEMATICAL REASONING EXPERIMENTS

**Model.** We employed the pre-trained Qwen2.5 checkpoints from Hugging Face Hub, released under the Apache 2.0 license: base models (Qwen/Qwen2.5-0.5B; Qwen/Qwen2.5-1.5B; Qwen/Qwen2.5-7B; Qwen/Qwen2.5-Math-7B) and instruction-tuned model (Qwen/Qwen2.5-Math-1.5B-Instruct).

**Dataset and evaluation.** For training, we used MetaMathQA (Yu et al., 2024) from Hugging Face Hub, which is released under the MIT license (meta-math/MetaMathQA). For evaluation, we used MathQA (Amini et al., 2019, Apache 2.0 license), GSM8K (Cobbe et al., 2021, MIT license), and MinervaMath (Lewkowycz et al., 2022, unspecified license), as provided in configurations from the Language Model Evaluation Harness (Gao et al., 2024) codebase, publicly available on GitHub under the MIT license (EleutherAI/lm-evaluation-harness). For the AMC23 benchmark, we utilized a dataset publicly available on the Hugging Face Hub (math-ai/amc23) under unspecified license.

**Optimization.** The setup is largely identical to the general-purpose instruction-following experiments described in Appendix B.2, with the only difference that the student model is not subjected to supervised fine-tuning prior to applying any distillation algorithms.

**Additional results for the 0.5B student.** We extend the results in Table 4 to the Qwen2.5-0.5B student. Unlike with larger students, i.e., 1.5B and 7B, the benefits of grafting are not observed for the 0.5B student. This suggests that grafting is effective only when the student has sufficient capacity, whereas for smaller students, purely mode-seeking approaches aligned with model compression remain more appropriate.

| Method | 0.5B ($p_\phi$) | | | | 1.5B ($p_\phi$) | | | | 7B ($p_\phi$) | | | |
|---|---|---|---|---|---|---|---|---|---|---|---|---|
| | MQA | GSM | MNV | Avg. | MQA | GSM | MNV | Avg. | MQA | GSM | MNV | Avg. |
| Teacher ($p_\theta$) | 45.0 | 27.3 | 42.0 | 38.1 | 45.0 | 27.3 | 42.0 | 38.1 | 45.0 | 27.3 | 42.0 | 38.1 |
| Student ($p_\phi$) | 28.5 | 5.2 | 14.2 | 16.0 | 35.0 | 8.8 | 27.1 | 23.6 | 43.4 | 11.3 | 39.1 | 31.3 |
| DistiLLM-2 | 30.1 | 33.5 | 15.4 | **26.3** | 37.4 | 40.9 | 30.1 | 36.1 | 47.9 | 46.7 | 40.1 | 44.9 |
| *w/ grafting (ours)* | 29.6 | 33.7 | 15.0 | 26.1 | 37.7 | 42.6 | 31.2 | **37.2** | 48.8 | 46.9 | 40.2 | **45.3** |

**Additional results for the 1.5B student.** We extend the analysis of the 1.5B student model results presented in Table 4 to include the reasoning-specialized model (deepseek-ai/DeepSeek-R1-Distill-Qwen-1.5B; MIT license; Guo et al., 2025), publicly available on Hugging Face Hub. The consistent improvements from our proposed grafting approach demonstrates, at a minimum, the potential scope of grafting to include more optimized, state-of-the-art reasoning models.

| Method | 1.5B ($p_\phi$) | | | | 1.5B-R1-Distill ($p_\phi$) | | | |
|---|---|---|---|---|---|---|---|---|
| | MQA | GSM | MNV | Avg. | MQA | GSM | MNV | Avg. |
| Teacher ($p_\theta$) | 45.0 | 27.3 | 42.0 | 38.1 | 45.0 | 27.3 | 42.0 | 38.1 |
| Student ($p_\phi$) | 35.0 | 8.8 | 27.1 | 23.6 | 37.8 | 17.5 | 37.9 | 31.1 |
| DistiLLM-2 | 37.4 | 40.9 | 30.1 | 36.1 | 40.9 | 42.4 | 38.5 | 40.6 |
| *w/ grafting (ours)* | 37.7 | 42.6 | 31.2 | **37.2** | 41.1 | 42.9 | 39.2 | **41.1** |

## B.4 CODE GENERATION EXPERIMENTS

**Model.** We employed the pre-trained DeepSeek-Coder checkpoints from Hugging Face Hub for our experiments, which are released under the DeepSeek Open Source License[3]: base models (deepseek-ai/deepseek-coder-1.3b-base; deepseek-ai/deepseek-coder-6.7b-base) and instruction-tuned models (deepseek-ai/deepseek-coder-1.3b-instruct; deepseek-ai/deepseek-coder-6.7b-instruct).

**Dataset and evaluation.** For training, we used the Evol-Instruct-Code-80k-v1 dataset from the Hugging Face Hub, which is released under the CC BY-NC-SA 4.0 license (nickrosh/Evol-Instruct-Code-80k-v1). The accompanying code for dataset generation is publicly available on GitHub under the Apache 2.0 license (nickrosh/evol-teacher; Roshdieh, 2023). This corresponds to the open-source implementation of Code Evol-Instruct (Luo et al., 2023). For evaluation, we used Hu-manEval (Chen et al., 2021, MIT license) and MBPP (Austin et al., 2021, Apache-2.0 license), both in their filtered form from the EvalPlus (Liu et al., 2023) codebase, publicly available on GitHub under the Apache-2.0 license (evalplus/evalplus).

**Optimization.** The setup is largely identical to the general-purpose instruction-following experiments described in Appendix B.2, with the only difference that the student model is not subjected to supervised fine-tuning prior to applying any distillation algorithms.

## C THE USE OF LARGE LANGUAGE MODELS

Large language models were used exclusively for polishing the writing, without contributing to the research ideas or content.

---

[3]https://deepseeklicense.github.io/

