# OpenReview forum: "From Compression to Generalization: Language Model Distillation With Grafting"
_ICLR.cc/2026/Conference — Submitted to ICLR 2026_

### Official Review · Reviewer_Tdbg · 2025-10-28

**Soundness:** 3
**Presentation:** 3
**Contribution:** 2
**Rating:** 6
**Confidence:** 4

**Summary:**

To address a typical trade-off in Large Language Model knowledge distillation—the balance between "mode-seeking" and "mode-covering", this paper proposes "grafting" a simple and easy-to-implement strategy. The method is shown to improve the student model's overall performance across various tasks, models, and algorithms.

**Strengths:**

1. **Clear and explicit motivation:** The paper strongly argues that as the goal of knowledge distillation shifts from mere model compression to broader generalization, a purely "mode-seeking" approach is insufficient and may even limit the model's generalization capacity.
2. **Sufficient experimental validation:** The experiments cover diverse tasks (e.g., instruction-following, math, code ), models (e.g., Qwen2.5, Llama3, DeepSeek-Coder ), and algorithms (e.g., SeqKD, GKD, DistiLLM-2 ), providing strong support for the central claims.
3. **Simple and easy-to-implement method:** The proposed "grafting" strategy is simple and intuitive. It combines sequence trees generated at multiple temperatures into a unified distillation target , thereby attempting to balance "mode-seeking" and "mode-covering" behaviors.

**Weaknesses:**

1. **Relatively limited performance gains:** While the performance improvements from "grafting" are consistent , the absolute gains are not highly significant in many cases. Even though the authors mention that sequence tree generation can be efficient (compatible with vLLM ), the limited gains might make practitioners using SOTA methods hesitate to introduce the extra workload and resource cost.
2. **Unclear hyperparameter selection:** The method introduces new hyperparameters, namely the set of temperatures $\mathcal{T}$ and the mixing distribution $\pi_0$ . The paper appears to default to a uniform distribution ($\pi_0(\tau) = 1/|\mathcal{T}|$ ) and a fixed set of temperatures (e.g., $\mathcal{T}=\{0.7, 1.0, 1.5, 2.0\}$ ). The paper lacks a discussion on how to select these parameters.
3. **Insufficient theoretical analysis:** The core idea of "grafting", mixing probabilities from sequence trees generated at multiple temperatures, appears to be a heuristic approach. The paper does not provide a deep theoretical analysis as to why this specific mixing strategy (Equation 8) is the optimal way to balance the two modes.

**Questions:**

1. Considering the additional complexity that 'Grafting' introduces by requiring sequence generation at multiple temperatures, could the authors please comment on the practical significance of these gains? That is, under what circumstances do you believe the benefits of introducing the strategy proposed in the paper outweigh the additional costs?
2. How sensitive is the method to the choice of $\mathcal{T}$ (e.g., the number of temperatures, their range, and intervals)? Did you experiment with non-uniform mixing distributions for $\pi_0$?
3. Have you considered dynamic or adaptive "grafting" strategies? For instance, could the mixing weights $\pi$ be adjusted based on the training step (e.g., as a form of curriculum) or even adapted based on the context?

---

> ### Author Response · Authors · 2025-11-17
>
> We sincerely thank the reviewer for the time and constructive feedback. We are particularly pleased that the reviewer found our motivation to be "clear," the methodology to be "simple" and "intuitive," and the experimental validation to be "sufficient" and "providing strong support" for our central claims. While the reviewer views our work positively, there are some remaining concerns regarding the design of our grafting approach. We believe that the detailed explanations provided in this response (and general response) will fully address these remaining issues.
>
> ---
>
> > (W1, Q1) While the performance improvements from "grafting" are consistent, the absolute gains are not highly significant ... Considering the additional complexity that 'Grafting' introduces ...
>
> While the reviewer notes that the improvement from grafting is "not highly significant," we wish to emphasize the value of the "consistent" improvements it provides, a point the reviewer also raised. Obtaining such consistent gains across a wide array of existing distillation algorithms (ranging from classical SeqKD and KD to more recent methods like GKD and DistiLLM-2), models, and various tasks, _all with virtually no additional cost (as described below)_, demonstrates that grafting is a highly compelling and attractive strategy with its "simple" and "intuitive" design.
>
> Crucially, our approach introduces virtually _no extra workload or resource cost_. The only requirements are utilizing multiple temperatures when generating the training sequence and appending a single column to the dataset indicating which temperature was used for that specific sequence. Please refer to the general response regarding this point.
>
> ---
>
> > (W2, Q2) The method introduces new hyperparameters, namely the set of temperatures ... How sensitive is the method to the choice of ...
>
> For this point as well, please refer to the general response. The bottom line is that our grafting approach also requires the use of generally acceptable temperature levels, and the values adopted in our experiments (0.7, 1.0, and 1.5) are a reasonable choice based on common practice, rather than the product of complex hyperparameter tuning.
>
> ---
>
> > (W2, Q2, Q3) The paper appears to default to a uniform distribution ... non-uniform mixing distributions ... Have you considered dynamic or adaptive ...
>
> We defaulted to a uniform prior $\pi_{0}$ for both theoretical and practical reasons. At the initial step $t=0$, we lack any sequence information, and thus a uniform distribution is the most natural and uninformative prior to use in this state. While an engineered, non-uniform prior might exist, this is not a principled prior and adds engineering complexity to the algorithm.
>
> This leads directly to your question on dynamic or adaptive strategies. Our formulation is, by design, an adaptive grafting strategy for $t > 1$. As defined in Equation (11), the mixing weights $\pi_{t}$ are dynamically updated at every generation step $t$. This $\pi_{t}$ is the posterior distribution, conditioned on the previously generated context $s_{0:t-1}$. Therefore, our framework already implements this core idea: we start with a simple, principled uniform prior ($\pi_{0}$) and then employ a dynamic, context-aware update to derive the posterior ($\pi_{t}$), which guides the grafting process at each subsequent step.
>
> ---
>
> > (W3) The paper does not provide a deep theoretical analysis ...
>
> Our paper is fundamentally experimental in nature. A theoretical analysis for knowledge distillation in deep neural networks remains inherently limited, and this restriction is amplified in application-intensive domains such as large language model distillation, where the various components required for practical usage make formal theoretical grounding challenging. Like many other machine learning algorithms, our grafting technique was developed from a rational conceptual basis rather than rigorous theory.
>
> Nonetheless, we emphasize that we have provided proper theoretical justification, at least for the practical implementation details of our approach (cf. Section 4.3 and Appendix A). Also, we do not claim that the uniform mixture in Equation (8) is theoretically optimal. Rather, it serves as the simplest and most principled uninformative prior for grafting the subtrees, as we discussed above. We encourage the reader to view our work not as a fully theoretically grounded framework, but rather as a method that leverages theory as a tool for creating a conceptually sound and practically useful approach.

---

> > ### Comment · Reviewer_Tdbg · 2025-11-22
> >
> > Thank you for the reply. I have carefully reviewed all the reviews and responses.

---

> > > ### Author Response · Authors · 2025-11-24
> > >
> > > Thank you for your response. We have uploaded the first revision incorporating the raised concerns, and we hope it meets your expectations.

---

### Official Review · Reviewer_wztp · 2025-10-30

**Soundness:** 3
**Presentation:** 3
**Contribution:** 3
**Rating:** 6
**Confidence:** 3

**Summary:**

This paper is concerned with the balance between the mode-seeking and mode-covering in knowledge distillation. It proposes a grafting strategy that integrates samples generated with diverse temperatures from the teacher LM into several existing knowledge distillation objectives (e.g., SeqKD). With this simple and effective strategy, the performance of the student LM is largely improved in both strong-to-weak and weak-to-strong scenarios.

**Strengths:**

1. The proposed strategy is rather simple and universally applicable to existing knowledge distillation methods.
2. The experiments involving both strong-to-weak and weak-to-strong cases are solid and show the strategy is promising.

**Weaknesses:**

1. The proposed strategy is mostly observation-drive and lacks a proper theoretical justification.
2. The experiments are mostly constrained to comparably smaller-scale LMs, and larger-scale LMs with tens of billions of parameters should also be considered. Similarly, LMs with different architectures (e.g., MoE LMs) would further strengthen the usefulness of the strategy.

**Questions:**

N/A

---

> ### Author Response · Authors · 2025-11-17
>
> We sincerely thank the reviewer for the time and positive feedback on our work. We are pleased that the reviewer accurately identified our method as "simple" and "universally applicable," and offered the promising assessment that it is "promising." We are also happy to hear the reviewer enjoyed our "solid" experiments. We hope that our detailed responses have fully addressed the concerns you raised.
>
> ---
>
> > (W1) The proposed strategy is mostly observation-driven ...
>
> Our paper is fundamentally experimental in nature. A theoretical analysis for knowledge distillation in deep neural networks remains inherently limited, and this restriction is amplified in application-intensive domains such as large language model distillation, where the various components required for practical usage make formal theoretical grounding challenging. Like many other machine learning algorithms, our grafting technique was developed from a rational conceptual basis rather than rigorous theory. Nonetheless, we emphasize that we have provided "proper" theoretical justification, at least for the practical implementation details of our approach (cf. Section 4.3 and Appendix A). This theoretical grounding, in turn, justifies the use of the surrogate loss $\hat{\mathcal{L}}$, which circumvents the need for explicit or expensive computations involving the grafted target, and forms the foundation for creating a "simple" and "universally applicable" practical algorithm.
>
> ---
>
> > (W2) The experiments are mostly constrained to comparably smaller-scale LMs, ...
>
> We appreciate the suggestion regarding broader empirical validation on "larger-scale" language models, scaled to the tens of billions of parameters. While such extensive experimentation is desirable, the required computational cost currently exceeds our available resources. Nonetheless, we wish to clearly state that we made every effort to verify scalability within our constraints; we conducted experiments on a 32B model, the results of which are presented in the "Scaling to larger student" paragraph (Lines 410-420) and summarized in Table 2. We believe our experimental validations, up to the 32B scale, have been conducted at a level sufficient for the academic community, in the "solid" manner.

---

### Official Review · Reviewer_ZV4z · 2025-11-02

**Soundness:** 3
**Presentation:** 3
**Contribution:** 3
**Rating:** 4
**Confidence:** 4

**Summary:**

This paper investigates the knowledge distillation of generative language models. The authors propose a core argument: traditional distillation methods focus excessively on "mode-seeking" behavior (i.e., high-fidelity imitation of the teacher), which, while suitable for model compression, impairs the ability of the student model (especially large-capacity ones) to capture generative diversity, thus limiting mode-covering generalization.
To address this tension, the paper introduces a new technique called "Grafting." This method combines the sequence trees (or probability distributions) generated by the teacher model at multiple different decoding temperatures into a single, more diverse distillation target, thereby achieving a better balance between mode-seeking and mode-covering.
The authors theoretically justify the soundness of this approach (e.g., providing a gradient equivalence proof for forward KL and a variational upper bound for reverse KL). Experimental results show that the Grafting strategy can consistently improve the performance of various existing distillation baselines (such as GKD, DistiLLM) on general-purpose instruction-following, mathematical reasoning, and code generation tasks. Notably, under weak-to-strong distillation setups, the student model's performance can even surpass that of the teacher.

**Strengths:**

1.Instead of viewing distillation merely as model compression, the paper offers a novel perspective that elevates the problem to a fundamental trade-off between "mode-seeking" (precision) and "mode-covering" (generalization/recall). This framework is significant for understanding and improving the distillation of generative models, particularly large-capacity ones.
2.The "Grafting" strategy is well-grounded in theory. The mathematical derivations in Appendix A are rigorous, proving that the proposed tractable loss is a valid proxy for the intractable objective. The gradient equivalence for the forward KL case is a particularly strong technical contribution.
3.The paper's analysis is highly mature. The fidelity analysis (Figure 5c) convincingly demonstrates that performance gains do not come from higher fidelity but from stronger generalization, which perfectly supports the "mode-covering" thesis. Furthermore, the authors are very candid in disclosing and analyzing a failure case (the 0.5B model in Appendix B.3). This counterexample, rather than being a weakness, strengthens the paper's core hypothesis by defining the method's scope of applicability (i.e., high-capacity student models).

**Weaknesses:**

While the paper's contributions are impressive, the strength of the core claims could be further reinforced by more in-depth ablation studies on implementation details and key hyperparameters.

**Questions:**

To more accurately assess the contribution of this work, I hope the authors will address or discuss the following questions in the final version:
1.The paper claims its method differs from "merely aggregating multi-temperature samples." Could the authors provide an experiment that fairly compares "Grafting" (sampling from the mixture distribution ) with "Data Aggregation" (mixing datasets sampled from each  individually) under the same total sample size? If their performance is similar, is the theoretical complexity of the "Grafting" approach (e.g., the proofs in Appendix A) still necessary?
2.The paper defaults to a uniform mixture in Eq. (8) (i.e., ). Why was this chosen? Given that Fig. 5a shows different single temperatures contribute differently (e.g.,  is strongest on Avg.), did the authors experiment with a weighted mixture? For instance, might a strategy weighted towards the optimal single temperature (e.g., 50% , 25% , 25% ) yield even better results than the uniform mixture?
3.Could the authors further explore the boundaries of this trade-off? If the "Grafting" strategy incorporates more or higher-temperature trees (e.g., ), is there a point of diminishing returns where the fidelity drops too low (excessive mode-covering), causing the final performance (e.g., Avg. winning rate) to decrease as well?

---

> ### Author Response · Authors · 2025-11-17
>
> We sincerely thank the reviewer for the time and constructive feedback. We are particularly pleased that the reviewer found our framework provides a "novel" perspective and makes a "significant" and "impressive" contribution, noting that it is "well-grounded in theory." Above all, we are truly delighted that the reviewer recognized the "highly mature" nature of our analysis and fully grasped our intention behind the "candid" discussion of the failure case. While the overall comment is positive, the reviewer has raised several key concerns aimed at further strengthening the paper. We are confident that these issues can be effectively addressed through the detailed explanations and additional results provided in this response.
>
> ---
>
> > (Q1) The paper claims its method differs from ...
>
> Figure 5(b) presents the results of that experiment, which is further described in the paragraph titled "Distilling with a grafted tree target" (Lines 393-399). Throughout the experiment, we maintained _"the same total sample size"_ for a fair comparison, and thus "S" and "S+T" in Figure 5(b) correspond directly to what you refer to as _"Grafting (sampling from the mixture distribution)"_ and _"Data Aggregation (mixing datasets sampled from each individually),"_ respectively. Clearly, the effect of "Grafting" certainly includes an element of "Data Aggregation," as you intuitively suggest, but using the grafted target results in a greater improvement. Furthermore, while the associated mathematical derivations might suggest _"theoretical complexity,"_ this complexity is primarily related to the derivation needed for the practical implementation of using the grafted tree as the distillation target. It is important to note that the actual implementation itself is not complex at all (Lines 320-323), as emphasized in the general response.
>
> ---
>
> > (Q2) The paper defaults to a uniform mixture in Eq. (8) ...
>
> A better weighting strategy for each subtree, beyond using a uniform distribution, would probably exist. However, implementing it would require introducing additional engineering factors, which would complicate the algorithm further. Moreover, Equation (11) shows that the weighting coefficient $\pi_t$ at the $t$-th generation step is conditioned on the previously generated sequence $s_{0:t-1}$. This implies that at the initial step, $\pi_0(\tau)$, a uniform prior is natural due to the lack of information, and the posterior update for $\pi_{t}$ is subsequently performed based on observations at time $t$. While assigning a non-uniform prior in this initial, uninformed state might be useful from an engineering perspective, it is not theoretically principled. For example, even a strategy based on the optimal temperature of each single-tree run, as you mentioned, is determined in a post-hoc manner and thus cannot be considered a proper prior $\pi_0$.
>
> ---
>
> > (Q3) Could the authors further explore the boundaries ...
>
> The temperature values of 0.7, 1.0, and 1.5 that we consider in our main experiments were set to a range that maintains a reasonable level of quality when using min-p sampling (Minh et al., 2025).
>
> In fact, in earlier experiments, we overestimated the ability of min-p sampling to "filter out less probable tokens" and used larger temperature values such as 3.0, which led to degradation in performance. Specifically, for the DistiLLM-2 setup (1.5B distilled from 3B-Instruct for general-purpose instruction-following) that yields an average winning rate of 46.1, using grafting up to temperature of 1.5 achieves a winning rate of 47.4 (cf. Table 1). However, including the higher temperature of 3.0 in the range reduces this performance to 46.6, thus demonstrating the importance of selecting an appropriate range of temperatures.
>
> Moreover, in additional benchmarks conducted on math tasks to address Reviewer FAkE's request, we observed a similar trend for the DistiLLM-2 setup (1.5B distilled from 1.5B-Instruct for mathematical reasoning). Using temperatures up to 1.5, 2.0, and 3.0 achieved average performance scores of 37.2, 36.7, and 36.6, respectively, progressively narrowing the performance gap with the baseline’s 36.1.
>
> These observations highlight that while composing the grafted target from multiple temperatures is beneficial, the selected range must be confined to generally acceptable temperature levels (e.g., values above 2.0 are typically not acceptable in practice).

---

### Official Review · Reviewer_FAkE · 2025-11-10

**Soundness:** 2
**Presentation:** 2
**Contribution:** 2
**Rating:** 4
**Confidence:** 2

**Summary:**

This study re-examines knowledge distillation for autoregressive generative language models, emphasizing that its core value extends beyond mere model compression. While pure mode-seeking methods are applicable for compression, they restrict student models’ ability to capture linguistic diversity and hinder generalization on language modeling tasks. To address this issue, the authors propose a concise "grafting strategy" that balances mode-seeking and mode-covering via fusing multi-temperature sequence trees, along with flexibility for both strong-to-weak and weak-to-strong distillation scenarios.

**Strengths:**

1. This study conducts experiments covering three key tasks—general instruction following, mathematical reasoning, and code generation—with three major model families (Qwen, Llama, and DeepSeek-Coder). It also explores two distinct distillation scenarios: distilling from stronger to weaker models and from weaker to stronger models, effectively validating the proposed method’s effectiveness.
2.  Knowledge distillation requires balancing mode-seeking and mode-covering. The proposed grafting strategy achieves this balance by combining multi-temperature sequence trees, thereby enhancing the model’s generalization ability.
3.  The paper provides valuable insights into the relationship between student model capacity and grafting gains: stronger student models yield more significant benefits from the grafting strategy, while insufficient student capacity leads to limited or even no gains.

**Weaknesses:**

1. The authors have not provided open-source code for the proposed method. This lack of reproducibility hinders other researchers from verifying the experimental results, building upon the work, or comparing it with alternative approaches—an essential practice in academic research.
2. The validation scope across datasets is relatively limited. For instance, in the mathematical reasoning task, only two datasets (MathQA and GSM8K) are evaluated, while other widely adopted benchmarks such as AMC, AIME, MinervaMath, and OlympiadBench are not included. A similar gap exists for the other tasks (general instruction following and code generation), where the method’s performance on a broader set of standard datasets remains unvalidated. This limits the generalizability of the reported findings.
3. For the mathematical reasoning and code generation tasks, the proposed method has not been evaluated on state-of-the-art reasoning-specialized models (e.g., models optimized for step-by-step reasoning or domain-specific logical deduction). Given the growing relevance of such models for these tasks, this omission makes it difficult to assess the method’s competitiveness in real-world scenarios where reasoning-capable models are commonly used.

**Questions:**

1. Please supplement experiments on the broader datasets mentioned in the Weaknesses section (e.g., AMC, AIME for math reasoning) to verify the grafting strategy’s generalizability.
2. Please evaluate the method on currently popular reasoning-specialized models and demonstrate its effectiveness on these reasoning-focused architectures.
3. Under strict data efficiency constraints (e.g., 1/10 or less of conventional training data), might multi-temperature sample aggregation outperform your grafting strategy in the performance-efficiency trade-off by avoiding complex probability fusion overhead? Do your conclusions only hold with sufficient data, and can you supplement low-data regime experiments to clarify this?

---

> ### Author Response · Authors · 2025-11-17
>
> We appreciate the reviewer for the time and constructive feedback. We are particularly pleased that the reviewer found our work provides "valuable insights" with comprehensive experiments covering "key tasks" with "major model families." We notice that the reviewer raised several concerns regarding the empirical validation, and we hope that our detailed responses and the additional experimental results provided will effectively address these issues.
>
> ---
>
> > (W1) The authors have not provided open-source code ...
>
> We fully recognize the crucial value of open-sourcing for the benefit of the academic community, particularly concerning reproducibility and future research. We intend to publicly release the complete code implementation of our approach alongside the camera-ready revision of the paper. However, we must address a potential licensing concern: our main experimental code is built upon the DistiLLM-2 codebase, which is currently under an unspecified license. Thus, we will first resolve this issue by contacting the original authors to ensure that our derived work can be released under an appropriate open-source license.
>
> ---
>
> > (W2, Q1) The validation scope across datasets is relatively limited. For instance, ... Please supplement experiments on the broader datasets mentioned in the Weaknesses section ...
>
> Additional experimental validation would certainly strengthen the paper. Following the reviewer’s suggestion, we conducted additional evaluations on the MinervaMath and AMC23 datasets. The results demonstrate that our proposed grafting approach continues to outperform the baseline, which further validates its generalizability. We will integrate these additional benchmark results into the final revised manuscript in an appropriate form.
>
> __Additional results for MinervaMath.__
> | Method | 1.5B | 7B | 7B-Math |
> | :- | :-: | :-: | :-: |
> | Teacher | 42.0 | 42.0 | 42.0 |
> | Student | 27.1 | 39.1 | 45.8 |
> | DistiLLM-2  | 30.1 | 40.1 | 45.6 |
> | w/ grafting | __31.2__ | __40.2__ | __46.3__ |
>
> __Additional results for AMC23.__
> | Method | 1.5B | 7B | 7B-Math |
> | :- | -: | -: | -: |
> | Teacher | 12 / 40 | 12 / 40 | 12 / 40 |
> | Student | 1 / 40 | 6 / 40 | 12 / 40 |
> | DistiLLM-2  | 10 / 40 | 18 / 40 | 12 / 40 |
> | w/ grafting | __11 / 40__ | __21 / 40__ | __14 / 40__ |
>
> ---
>
> > (W3, Q2) For the mathematical reasoning and code generation tasks, the proposed method has not been evaluated on state-of-the-art reasoning-specialized models ... Please evaluate the method on currently popular reasoning-specialized models ...
>
> We kindly ask the reviewer to recognize that, within the constraints of our computational resources, we made every effort to select the most up-to-date and highly specialized math and code models available. In particular, we employed the latest math-specialized model within Qwen's collection on the HuggingFace Hub: the Qwen2.5-Math family, rather than the Qwen2-Math family. While we were aware of the newer DeepSeekCoder-V2 in deepseek-ai's collection, we opted for its predecessor, DeepSeek-Coder. This decision was strictly due to resource constraints, as DeepSeekCoder-V2 is configured in sizes (16B and 236B) that exceed our available computational capacity. Given these efforts, we believe we have conducted appropriate validation against models already optimized for these domains. In light of this, we would respectfully appreciate it if the reviewer could further elaborate on the meaning of state-of-the-art reasoning-specialized models.
>
> ---
>
> > (Q3) Under strict data efficiency constraints ... might multi-temperature sample aggregation outperform your grafting strategy in the performance-efficiency trade-off by avoiding complex probability fusion overhead? ...
>
> We thank the reviewer for this intriguing question regarding data efficiency. However, we wish to clarify a potential misunderstanding regarding the computational cost of our method and the associated concern about "complex probability fusion overhead." As detailed in Section 4.3, our grafting strategy is explicitly designed to avoid complex or computationally expensive fusion processes. In practice, the procedure is simple: we generate training sequences using multiple temperatures and append a single column to the dataset indicating the temperature used for each sequence (note that we ensure a fair comparison by maintaining the exact same total number of training sequences throughout the experiments). The "fusion" is handled implicitly during loss computation via the surrogate loss $\hat{\mathcal{L}}$, which replaced the standard loss $\mathcal{L}$. Crucially, thanks to the mathematical derivation provided in Appendix A, this formulation circumvents the need for explicit or expensive computations involving the "complex" $q_{\theta}$. Consequently, even under strict data or resource constraints, there is virtually no "efficiency trade-off," as removing the grafting mechanism would yield negligible computational savings.

---

### Author Response · Authors · 2025-11-17

We are sincerely grateful for all the reviewers’ dedicated time and their constructive feedback. While we are pleased to note that the contributions of our paper were generally evaluated positively, there appears to be a shared concern across the reviews regarding two specific points: computational costs and the rationale for the temperature setting. It would be nice to address these points comprehensively through this general response.

---

__Resolving a potential misunderstanding regarding computational costs.__

Reviewers FAkE and Tdbg raised concerns about the additional computational cost introduced by the grafting approach. We can confidently state that this cost is virtually nonexistent. Thanks to the use of the surrogate loss $\hat{\mathcal{L}}$ (justified in Section 4.3 and Appendix A), the only requirements for employing our grafting method in existing state-of-the-art distillation algorithms (such as GKD and DistiLLM-2) are generating the training data with varying temperatures and explicitly marking the temperature used for each generated sequence.

We would also like to emphasize that the total number of training sequences was kept consistent to ensure a fair comparison. Specifically, a reasonable and equitable comparison was maintained throughout the experiments, such as comparing generating 999 sequences at a single temperature versus generating the same total number (999 sequences) by sampling 333 sequences each at three different temperatures.

---

__Discussions on temperatures and prior weights.__

Reviewers ZV4z and Tdbg raised important questions regarding the setting of the temperature values and mixing weights, which are key components of our proposed grafting approach.

As Reviewer ZV4z accurately anticipated, employing an excessively large temperature leads to a degradation in performance. This observation aligns with common practice and knowledge in large language model generation, where high temperatures increase output entropy but often diminish overall quality. When this quality degradation becomes excessive—for example, in our experiments, we considered temperatures up to 3.0, a value that clearly exceeds the typical range (temperatures are generally not set above 2.0 in practice)—the effectiveness of our grafting technique is also reduced. This observed drop in efficacy would precisely represent the "point of diminishing returns" that Reviewer ZV4z referred to. Consequently, our grafting approach also requires the use of generally acceptable temperature levels, and the values adopted in our experiments (0.7, 1.0, and 1.5) are a reasonable choice based on common practice, rather than the product of complex hyperparameter tuning.

Next, Reviewers ZV4z and Tdbg both inquired about the potential of using a weighted or adaptive mixture instead of the uniform mixture we currently employ for $\pi_{0}$. This is an excellent segue into an intriguing discussion concerning the weighting coefficient ($\pi_{t}$) assigned to each individual subtree. In the formulation presented in our paper, $\pi_{0}$ can be viewed as the prior and $\pi_{t}$ as the posterior. Since $\pi_{0}$ reflects the state at time $t=0$ where we have no information regarding how good each temperature's subtree is, it is more rational to employ a noninformative uniform prior, even though an engineered weighted mixture might potentially yield superior results. Subsequently, we observe that $\pi_{t}$ is conditioned on the path traversed on the grafted sequence tree up to that point. This structure effectively constitutes a posterior update based on preceding observations from the initial uniform prior, rendering the mixture, in some sense, an adaptive one. Including a detailed discussion on this $\pi$ coefficient in the camera-ready revision will undoubtedly aid in understanding the mechanics of our grafting approach and will further solidify the paper. We are thankful for leading this constructive discussion.

---

In addition to these two points, the reviewers raised several other important comments. We hope that our individual responses to each reviewer address these concerns to their satisfaction.

Sincerely,
Authors of Submission1738

---

### Author Response · Authors · 2025-11-24

We are pleased to present the first revision to all of you. The modified and added content is marked in blue and primarily addresses the following key concerns:
- Addition of results for the _MinervaMath and AMC23 benchmarks_ within the mathematical reasoning experiments of Section 5.2.
- Discussion of _computational costs and mixture weights_ in the Remark paragraphs of Section 4.
- Explicit confirmation at the beginning of Section 5 that the experiments were conducted under _the same number of training sequences for fair comparison_.

We appreciate the constructive feedback once again, which has contributed to improving the quality of our paper. If there are any remaining concerns, please let us know.

Sincerely,
Authors of Submission1738

---

### Author Response · Authors · 2025-11-27

As we are now in the final third of the discussion phase, we have further revised our manuscript. The content that has been modified and added is marked in blue and responds to the following matters, building upon the first revision:

- We incorporated rigorous ablation results in Section 5.2 that fully address the concern regarding the _temperature range_ raised by Reviewers ZV4z and Tdbg. We summarized our findings by noting that a diminishing return was observed: as the temperature upper limit was increased (1.5 -> 2.0 -> 3.0), the average performance concurrently decreased (37.2 -> 36.5 -> 36.2).
- We added further experimental results using the _R1-Distill model_ (detailed in Appendix B.3). Although it does not directly utilize the state-of-the-art reasoning models with tens of billions of parameters mentioned by Reviewers FAkE and wztp, we believe it minimally demonstrates the potential scope of our grafting method to include such highly reasoning-specialized models.

We appreciate the constructive feedback once again, which has contributed to improving the quality of our paper. If there are any remaining concerns, please let us know.

Sincerely,
Authors of Submission1738

---

### Author Response · Authors · 2025-11-30

We are deeply grateful to the entire Program Committee for their valuable time and dedication. To facilitate a smooth review process, we now summarize our final revisions as action items, detailing precisely how we have addressed the reviewers' specific concerns.

Overall, independent of the initial ratings (4, 4, 6, 6), the initial feedback provided by the reviewers was already quite positive:

- Reviewer FAkE (4) found our work provides _"valuable insights"_ with comprehensive experiments covering _"key tasks"_ with _"major model families."_
- Reviewer ZV4z (4) found our framework provides a _"novel"_ perspective and makes a _"significant"_ and _"impressive"_ contribution, noting that it is _"well-grounded in theory"_ and provides _"highly mature"_ analysis.
- Reviewer wztp (6) accurately identified our method as _"simple"_ and _"universally applicable,"_ and noted that it is _"promising."_
- Reviewer Tdbg (6) found our motivation to be _"clear,"_ the methodology to be _"simple"_ and _"intuitive,"_ and the experimental validation to be _"sufficient"_ and _"providing strong support"_ for our central claims.

The initial scores likely reflected the specific concerns raised by the reviewers. __*We are confident that the following action items successfully resolve all major issues*__, and despite the lack of further replies from reviewers other than Reviewer Tdbg within the response period, our revisions are expected to have already secured a positive change in the final ratings.

---

__(Action Item #1) Clarifying Computational Costs.__

- We addressed the misunderstanding raised by Reviewers FAkE and Tdbg regarding the potential additional computational cost of the grafting approach by clearly demonstrating that __*no such additional cost exists*__.
- Reviewer Tdbg confirmed this clarification in a subsequent reply made within the response period.
- We have revised the "Remark" paragraph in the original manuscript and retitled it as __*"Remark on computational costs (line 323)"*__ to make it more easily understandable for the reader.

---

__(Action Item #2) Explaining Mixture Weights.__

- We addressed the question raised by Reviewers ZV4z and Tdbg regarding the uniform mixture setup of the grafting approach by clearly demonstrating that __*the uniform weighting at t=0 is both a principled prior and serves to avoid complex hyperparameter engineering; moreover, it naturally yields a non-uniform mixture for t>0*__.
- Reviewer Tdbg confirmed this explanation in a subsequent reply made within the response period.
- We have added the paragraph __*"Remark on mixture weights (line 345)"*__ to enhance the reader's understanding of the mechanism and operation of the grafting approach.

---

__(Action Item #3) Investigating Temperatures.__

- We addressed the question raised by Reviewers ZV4z and Tdbg regarding the appropriate temperature setup of the grafting approach by clearly demonstrating that __*excessively high temperatures can lead to a degradation in performance*__.
- Reviewer Tdbg confirmed this point in a subsequent reply made within the response period.
- We have added the paragraph __*"Temperature range and diminishing returns (line 497)"*__ and __*"Table 7"*__ to show that the grafting approach functions effectively within a commonly used temperature range.

---

__(Action Item #4) Additional Benchmarking Tasks.__

- We addressed the demand raised by Reviewer FAkE regarding additional benchmark results for mathematical reasoning tasks by further demonstrating experimental results showing that grafting consistently improves performance on __*the MinervaMath and AMC23 benchmarks*__.
- We have revised __*"Table 4"*__ to include MinervaMath results and added a new __*"Table 6"*__ to include AMC23 results.

---

__(Action Item #5) Additional Benchmarking Models.__

- We addressed the demand raised by Reviewers FAkE and wztp regarding additional benchmark results for state-of-the-art reasoning models by further demonstrating experimental results showing that grafting consistently improves performance on __*the DeepSeek-R1 distilled model*__.
- We have revised __*"Appendix B.3 (line 1021)"*__ to include the results. While we could not utilize the DeepSeek-R1 model with 671 billion parameters due to computational constraints, our results with the R1-Distill model sufficiently indicate the potential scope of our grafting method on such highly reasoning-specialized models.

---

We once again express our sincere gratitude for the efforts of all the reviewers and chairs, and hope this response facilitates the final decision.

Sincerely,
Authors of Submission1738

---

### Meta-Review · Area_Chair_5qy2 · 2025-12-16

**Summary:**

This paper focuses on the "trade-off between mode-seeking and mode-covering" in language model distillation and proposes *grafting*, a method to integrate sequence trees generated at multiple temperatures into a single distillation target. Its strength lies in its design, which is relatively easy to incorporate into existing distillation objectives such as SeqKD, GKD, and DistiLLM. The paper reports generally consistent improvements across multiple tasks (instruction-following, math reasoning, code generation), multiple model families, and both strong-to-weak and weak-to-strong settings.

On the other hand, the primary points of contention regarding acceptance were: (i) the margin of improvement is often small in many settings, making the practical cost-effectiveness (whether it justifies the extra effort) borderline; (ii) additional ablation studies regarding design guidelines/sensitivity for temperatures and mixture weights cannot be said to be fully sufficient; (iii) reviewers are divided on the theoretical positioning (where principled arguments end and empirical findings begin); and (iv) reproducibility (code release) remains unresolved at present.

**Reviewer Concerns:**

- **Computational Cost / Practical Additional Burden** (FAkE, Tdbg): Concerns were raised about whether multi-temperature generation adds excessive cost. The authors explain that the additional burden during training is effectively small or negligible due to "data generation at multiple temperatures + temperature labeling" and "avoiding explicit probability fusion via surrogate loss."
- **Difference between grafting and simple data aggregation** (ZV4z): Comparison under the same total sample count. The authors pointed to existing comparative experiments (explanation corresponding to Fig. 5(b)) to clarify that grafting outperforms simple aggregation.
- **Validity/Sensitivity of Temperature Range and Mixture Weights, Additional Ablations** (ZV4z, Tdbg): Questions regarding the basis for uniform weights, degradation when temperature is raised too high (diminishing returns), and lack of design guidelines. The authors position uniform weights as a "natural prior in an uninformed initial state" and explain that it becomes effectively adaptive through posterior updates during the generation process. They also provided additional observations showing performance drops at high temperatures. However, a more systematic sensitivity analysis (exploration of temperature sets/weights and general recommendations) remains as future work.
- **Positioning of Theoretical Claims** (wztp, Tdbg vs. ZV4z): Evaluations are split; some reviewers assess the theoretical backing as insufficient or heuristic, while another reviewer evaluates the derivation in the Appendix as a strong contribution.
- **Reproducibility (Code Release) and Evaluation Scope** (FAkE, wztp): Code has not been released yet, and there is insufficient verification on datasets or larger/different architectures. The authors stated they plan to release the code after checking the codebase license and presented additional benchmarks (e.g., MinervaMath/AMC23).

**Reviewer Scores:**

- **Reviewer FAkE (Initial 4, confidence 2):** While the authors presented additional benchmarks and cleared up misunderstandings regarding cost, some requests (e.g., broader SOTA reasoning models) were not met. **Most likely maintaining 4 -> 4.**
- **Reviewer ZV4z (Initial 4, confidence 4):** The content itself is highly rated; the main deficiencies were ablation/hyperparameter design and clarification of the difference from aggregation. Since the authors presented the positioning of the comparison with the same total sample count (Fig. 5(b)) and explanations/additional observations regarding temperature/weights, **the possibility of rising from 4 -> 6 is relatively high.**
- **Reviewer wztp (Initial 6, confidence 3):** Appreciates the "simplicity and ease of application" and "solid experiments," but has requests regarding theory/scale. Even considering the response, the basic stance does not seem to change significantly, **maintaining 6 -> 6.**
- **Reviewer Tdbg (Initial 6, confidence 4):** Concerned about the small margin of improvement, hyperparameter selection, and depth of theory. The authors reinforced points about the low additional cost and the prior/posterior explanation, **maintaining 6 -> 6.**

Predicted Average: (4 + 6 + 6 + 6) / 4 = **5.5**.

---

### Decision · Program_Chairs · 2026-01-26

Reject